# Biologically Plausible Brain Graph Transformer

**Ciyuan Peng**[1], **Yuelong Huang**[2], **Qichao Dong**[3], **Shuo Yu**[2], **Feng Xia**[4]*, **Chengqi Zhang**[5], **Yaochu Jin**[6]*

[1]Federation University Australia, [2]Dalian University of Technology, [3]Zhejiang Gongshang University,
[4]RMIT University, [5]Hong Kong Polytechnic University, [6]Westlake University

## Abstract

State-of-the-art brain graph analysis methods fail to fully encode the small-world architecture of brain graphs (accompanied by the presence of hubs and functional modules), and therefore lack biological plausibility to some extent. This limitation hinders their ability to accurately represent the brain's structural and functional properties, thereby restricting the effectiveness of machine learning models in tasks such as brain disorder detection. In this work, we propose a novel Biologically Plausible Brain Graph Transformer (BioBGT) that encodes the small-world architecture inherent in brain graphs. Specifically, we present a network entanglement-based node importance encoding technique that captures the structural importance of nodes in *global information propagation* during brain graph communication, highlighting the biological properties of the brain structure. Furthermore, we introduce a functional module-aware self-attention to preserve the *functional segregation and integration* characteristics of brain graphs in the learned representations. Experimental results on three benchmark datasets demonstrate that BioBGT outperforms state-of-the-art models, enhancing biologically plausible brain graph representations for various brain graph analytical tasks[1].

## 1 Introduction

Brain graphs, also known as brain networks, are a primary form to present the complex interactions among regional activities, functional correlations, and structural connections within the brain (Seguin et al., 2023; Wu et al., 2024b; Zhu et al., 2024). Brain graphs are constructed based on information extracted from brain data, such as functional magnetic resonance imaging (fMRI), with regions of interest (ROIs) as nodes and the correlations among ROIs as edges. One of the most important characteristics of brain

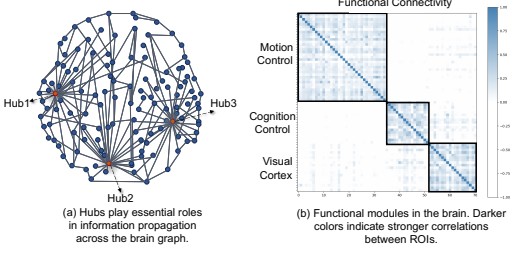

(a) Hubs play essential roles in information propagation across the brain graph.

(b) Functional modules in the brain. Darker colors indicate stronger correlations between ROIs.

Figure 1: Small-world architecture of brain graphs.

graphs is their *small-world architecture*, with scientific evidence supporting the presence of *hubs* and *functional modules* in brain graphs (Liao et al., 2017; Swanson et al., 2024). First, it is demonstrated that nodes in brain graphs exhibit a high degree of difference in their importance, with certain nodes having more central roles in information propagation (Lynn & Bassett, 2019; Betzel et al., 2024). These nodes are perceived as hubs, as shown in Figure 1 (a) (the visualization is based on findings by Seguin et al. (2023)), which are usually highly connected so as to support efficient communication within the brain. Second, human brain consists of various functional modules (e.g., visual cortex), where ROIs within the same module exhibit high functional coherence, termed functional integration, while ROIs from different modules show lower functional coherence, termed functional segregation (Rubinov & Sporns, 2010; Seguin et al., 2022). Therefore, brain graphs are characterized by community structure, reflecting functional modules. Figure 1 (b) visualizes the functional connectivity of a sample brain from ADHD-200[2] dataset. The functional module labels are empirically

---

*Corresponding authors: f.xia@ieee.org, jinyaochu@westlake.edu.cn

[1]Our code is available at https://github.com/pcyyyy/BioBGT.

[2]https://fcon_1000.projects.nitrc.org/indi/adhd200/

provided based on Dosenbach et al. (2010). ROIs in the same module have strong connections (high temporal correlations), while those from different modules show weaker connections.

With the significant ability of graph transformers in capturing interactions between nodes (Ma et al., 2023a; Shehzad et al., 2024; Yi et al., 2024), Transformer-based brain graph learning methods have gained prominence (Kan et al., 2022; Bannadabhavi et al., 2023). Despite these advancements, there is still a lack of tailored design for brain graphs that consider the small-world architecture as an essential feature. Consequently, the learned representations of current methods may lack sufficient biological plausibility for brain graphs. This limitation can be understood from two perspectives. First, most current studies consider connections within the brain as pairwise correlations between nodes, and typically treat all nodes equally. For example, Brain Network Transformer (Kan et al., 2022) assumes that all nodes in a brain graph have the same degree and each node is connected with all the other nodes. However, nodes can have significantly different roles in terms of propagating information within the brain graph (Lynn & Bassett, 2019). Second, current methods often encode the correlations between nodes simply based on node-level similarities, ignoring the existence of functional modules within a brain graph. Unfortunately, the existing labeling of functional modules is largely empirical and lacks precision (Kan et al., 2022). Therefore, this limitation is particularly evident in brain datasets where functional module labels are unavailable or inaccurate. This impedes preserving the functional segregation and integration characteristics of the brain.

To this end, this paper proposes a brain graph representation learning technique that departs from existing methods. We aim to improve the alignment of the learned representations with biological properties, particularly by encoding small-world features commonly observed in brain graphs. We propose a Biologically Plausible Brain Graph Transformer (BioBGT), which aligns brain graph representations with biological properties through two main components: *node importance encoding* and *functional module encoding*. (**1**) For brain graphs as communication networks, node importance is reflected by how crucial a node is in propagating information across the network (Seguin et al., 2023). Thus, we propose a node importance encoding technique based on *network entanglement*. Given the topology of a brain graph, the global information diffusion process is modeled through quantum entanglement, wherein the importance of a node is measured by the changes in the density matrix-based spectral entropy before and after perturbing the local connections surrounding the node. The encoding of node importance is thereafter embedded into node representations, reflecting the small-world architecture in terms of the presence of hubs. (**2**) We then present a functional module-aware self-attention to preserve the functional segregation and integration characteristics of brain graphs in the learned representations. Particularly, we design a community contrastive strategy-based functional module extractor to refine nodes' similarities at the functional modular level, instead of merely calculating node correlations at the node level. Therefore, we can obtain *functional module-aware node representations* for the self-attention mechanism.

**Contributions.** This paper highlights that brain graph representations obtained from learning models should align closely with the biological properties of the brain. Under this perspective, **i**) we propose a new Biologically Plausible Brain Graph Transformer entitled BioBGT that encodes the small-world architecture of brain graphs to enhance the biological plausibility of the learned representations; **ii**) we present a network entanglement-based node importance encoding technique, capturing node importance in the information propagation across brain graphs; **iii**) we introduce a functional module-aware self-attention, yielding functional module-aware node representations with the functional segregation and integration characteristics of brain graphs preserved; **iv**) experimental results show the effectiveness of our model design and the superiority of our model performance, especially in brain disease detection tasks.

## 2 PRELIMINARIES

### 2.1 PROBLEM DEFINITION

A brain graph presents the connectivity between ROIs, characterized by the small-world architecture. A brain graph with $n$ nodes (ROIs) is denoted as $G = (V, E, \mathbf{X})$, where $V$ stands for the node set, $E$ is the edge set, and $\mathbf{X} \in \mathbb{R}^{n \times d}$ represents the feature matrix with the $i$-th row vector $\mathbf{x}_i \in \mathbb{R}^d$ indicating the feature of node $i$. Here, $d$ is the hidden feature dimension. Hubs and functional modules are two crucial indicators of the small-world brain graph (Rubinov & Sporns, 2010). This paper suggests that the biological plausibility of brain graph representations can be enhanced by

capturing these features to an extent. For a given brain graph, our model aims to learn representations that incorporate biologically relevant features and achieve accurate brain graph analysis.

## 2.2 Graph Transformers

A Transformer architecture is composed of multiple Transformer layers, each of which contains a self-attention module followed by a feed-forward network (FFN) (Vaswani et al., 2017). In the self-attention module, the input feature matrix $\mathbf{X} \in \mathbb{R}^{n \times d}$ is first projected to query matrix $\mathbf{Q}$, key matrix $\mathbf{K}$, and value matrix $\mathbf{V}$ by the corresponding projection matrices $\mathbf{W}_Q \in \mathbb{R}^{d \times d_{\mathcal{K}}}$, $\mathbf{W}_K \in \mathbb{R}^{d \times d_{\mathcal{K}}}$, and $\mathbf{W}_V \in \mathbb{R}^{d \times d_{\mathcal{K}}}$:

$$\mathbf{Q} = \mathbf{X}\mathbf{W}_Q, \quad \mathbf{K} = \mathbf{X}\mathbf{W}_K, \quad \mathbf{V} = \mathbf{X}\mathbf{W}_V. \tag{1}$$

Then, the self-attention is calculated as:

$$\mathbf{A} = \frac{\mathbf{Q}\mathbf{K}^{\mathsf{T}}}{\sqrt{d_{\mathcal{K}}}}, \quad Attn(\mathbf{X}) = softmax(\mathbf{A})\mathbf{V}. \tag{2}$$

Here, $\mathbf{A}$ indicates the attention matrix representing the similarity between queries and keys, $d_{\mathcal{K}}$ is the dimension of $\mathbf{Q}$, $\mathbf{K}$, and $\mathbf{V}$. Extending Equation (2) to the multi-head attention is common and straightforward. Afterwards, the output of the self-attention module is fed to a FFN module:

$$\tilde{\mathbf{X}} = \mathbf{X} + Attn(\mathbf{X}), \quad \hat{\mathbf{X}} = \mathbf{W}_2 ReLU(\mathbf{W}_1\tilde{\mathbf{X}}). \tag{3}$$

Here, $ReLU(\cdot)$ stands for the activation function. $\mathbf{W}_2$ and $\mathbf{W}_1$ are the projection matrices.

Graph transformers are proposed for applying Transformers to graph data, which introduces the structural information of graphs as structural encoding (SE) or positional encoding (PE), such as Laplacian PE, spatial encoding, and edge encoding (Dwivedi et al., 2022; Geisler et al., 2023; Deng et al., 2024; Xing et al., 2024). However, these methods exhibit limitations when applied to brain graphs because they do not adapt to the specific small-world characteristic, including the presence of hubs in information propagation and functional modules.

## 3 Biologically Plausible Brain Graph Transformer

In this section, we present BioBGT in detail. We describe how to enhance the biological plausibility of brain graph representations by focusing on two key aspects: node importance encoding and functional module encoding. First, we design a network entanglement-based node importance encoding method in the input layer, denoted as $\Phi(\cdot)$. Then, to encode functional modules, we present a functional module-aware self-attention, denoted as FM-$Attn(\cdot)$. Therefore, for each node, we rewrite the left part of Equation (3) as:

$$\tilde{\mathbf{x}}_i = \Phi(\mathbf{x}_i) + \text{FM-}Attn(i). \tag{4}$$

Figure 2 shows the overall framework of our model. We will introduce the functions of $\Phi(\cdot)$ and FM-$Attn(\cdot)$ in Section 3.1 and Section 3.2, respectively.

### 3.1 Network Entanglement-based Node Importance Encoding

We measure node importance in information propagation based on network entanglement, importing quantum entanglement into brain graphs. Quantum entanglement is a phenomenon in quantum mechanics, describing the correlations between particles (Yu et al., 2023). Mathematically, quantum entanglement is often represented by a density matrix of quantum entangled states, which captures the entangled relationships between particles in the entire entangled system (Weedbrook et al., 2012). When combined with network information theory, concepts from quantum entanglement can provide a powerful lens for analyzing the *global topology* and *information diffusion* of graphs (Huang et al., 2024). Inspired by this, we treat the brain graph as an entangled system, where nodes and their connections reflect interdependent states. The density matrix is used to quantify structural information. This approach enables us to capture the intricate entangled relations between nodes, offering insight into both the global topological features and the information diffusion process within brain graphs.

**Proposition 1** (Density matrix as structural information)**.** *The structural information of a brain graph G, including the connection strength between nodes and the degree distribution of nodes, is encoded by its density matrix, which stands as a normalized information diffusion propagator and formulated*

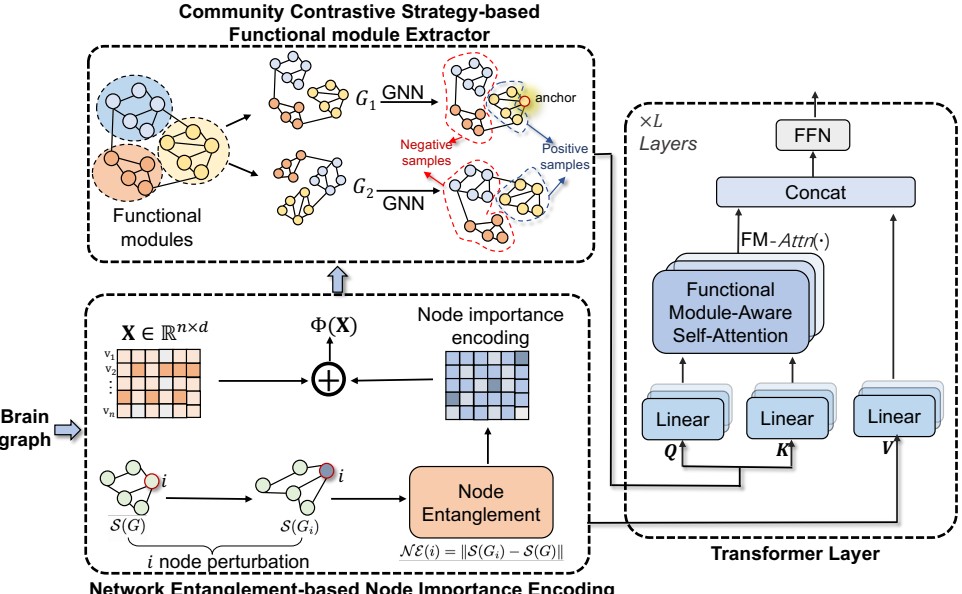

Figure 2: Overall framework of BioBGT.

as $\rho_G = \frac{e^{-\gamma \mathbf{L}}}{Z}$. Here, $e^{-\gamma \mathbf{L}}$ *is the information diffusion propagator,* $\gamma$ *denotes the positive parameter,* $\mathbf{L}$ *is the Laplacian matrix of* $G$, *and* $Z = Tr(e^{-\gamma \mathbf{L}})$ *represents the partition function of* $G$.

Appendix A.1 gives the complete proof. In quantum information theory, von Neumann entropy is used to measure the uncertainty or randomness of quantum systems (Huang et al., 2024). It quantifies the degree of entanglement present in quantum systems. When it comes to the complex graph scenario, density matrix-based spectral entropy is considered as the counterpart of von Neumann entropy, capturing the global topology and information diffusion process of graphs (De Domenico & Biamonte, 2016). It is formulated as:

$$\mathcal{S}(G) = -Tr(\rho_G \log_2 \rho_G), \tag{5}$$

where $\mathcal{S}(G)$ is the density matrix-based spectral entropy of $G$, and $Tr(\cdot)$ indicates the trace operation computing the trace of the product of the density matrix $\rho_G$ and its natural logarithm. The perturbation of a single node on the whole graph can be quantified by the change of density matrix-based spectral entropy, defined as node entanglement (NE) (Huang et al., 2024). We define node importance degree based on the NE value.

**Definition 1** (Node importance degree). *The node importance degree of node* $i$ *is defined as its NE value, formulated as* $\mathcal{NE}(i) = \|\mathcal{S}(G_i) - \mathcal{S}(G)\|$, *where* $G_i$ *denotes the* $i$-*control graph obtained after the perturbation of node* $i$. *A node with a higher NE value is considered more important and possesses hub attributes, exhibiting a greater disparity between the density matrix-based spectral entropy of the original graph and that of the perturbed graph.*

Notably, NE measures the node importance in terms of the influence of one node on the global topology and information diffusion throughout the graph. Compared to other methods, such as degree centrality (DC), betweenness centrality (BC), closeness centrality (CC), and eigenvector centrality (EC), which emphasize the local structure or local message passing, NE is more reliable for node importance measuring (especially in communication networks like brain graphs). Appendix B gives a detailed discussion. The next theorem shows the quantification analysis of entanglement.

**Theorem 1** (Quantification analysis of entanglement). *Assume that the number of connected components in the* $i$-*control graph is the same as the original graph, denoted as* $\alpha_i = \alpha$. *The NE value of node* $i$ *is approximated as*

$$\mathcal{NE}(i) \approx \left\| \frac{2m\gamma n^2}{\ln 2(n-\alpha)^2} \frac{\Delta Z}{Z Z_i} + \log_2(\frac{Z_i}{Z}) \right\|, \tag{6}$$

*where,* $n$ *and* $m$ *are the numbers of nodes and edges, respectively.* $Z_i$ *stands as the partition function for* $G_i$, *and* $\Delta Z = Z_i - Z$.

The complete proof is given in Appendix A.2. Then, we design our node importance encoding $\Phi(\cdot)$ by assigning each node the learnable embedding vector of its node importance degree in the input layer. For node $i$, its node representation in input layer is updated to $\mathbf{x}'_i$:

$$\mathbf{x}'_i = \Phi(\mathbf{x}_i) = \mathbf{x}_i + \mathbf{x}_{\mathcal{NE}(i)} \tag{7}$$

$\mathbf{x}_{\mathcal{NE}(i)}$ is the learnable embedding vector specified by $\mathcal{NE}(i)$.

## 3.2 FUNCTIONAL MODULE-AWARE SELF-ATTENTION

In this section, we first propose a community contrastive strategy-based functional module extractor, which can capture the functional segregation and integration characteristics of the brain. Then, the obtained functional module-aware node representations from the extractor are learned by an updated self-attention mechanism, which can calculate node similarity at the functional module level.

### 3.2.1 COMMUNITY CONTRASTIVE STRATEGY-BASED FUNCTIONAL MODULE EXTRACTOR

Given a brain graph $G$, the representation of node $i$ after node importance encoding is $\mathbf{x}'_i$, we then can obtain its updated representation after our functional module extractor $\psi$, indicated as $\mathbf{h}_i := \psi(i, \mathcal{M}_i)$, where $\mathcal{M}_i$ stands for the functional module node $i$ belongs to.

In $\psi$, we first utilize an unsupervised community detection method, Louvain algorithm (Blondel et al., 2008), to highlight the functional modules. This approach particularly addresses the challenge posed by the absence of functional module labels, which is a limitation encountered in many empirically labeled datasets. Then, we apply graph augmentation to generate two graph views of $G$ by modifying its structural information and enhancing functional modules. Particularly, we apply an edge dropping strategy (Rong et al., 2020; Chen et al., 2023) to achieve graph augmentation. The main idea of the edge dropping strategy is dropping less important edges while preserving the functional module structure. Details of edge dropping strategy are given in Appendix C. After graph augmentation, we can obtain two augmented graph views $G^1$ and $G^2$.

Then, $G^1$ and $G^2$ are fed into a graph neural network-based view encoder $\text{GNN}(\cdot)$ to obtain the representations of two graph views, denoted as $\mathbf{H}^1 \sim \text{GNN}(G^1)$ and $\mathbf{H}^2 \sim \text{GNN}(G^2)$. To enhance *inter-module differences* and *intra-module similarities*, we design a contrastive objective strategy by setting nodes from the same function module as positive samples, while those from different function modules as negative samples. We adopt the InfoNCE (Oord et al., 2018) as the contrastive loss function:

$$\mathcal{L} = -\frac{1}{n} \sum_{i=1}^{n} \log \frac{exp(Sim(\mathbf{h}_i^1, \mathbf{h}_i^{pos}))}{\sum_{j=1}^{n^{\text{Neg}}} exp(Sim(\mathbf{h}_i^1, \mathbf{h}_j^1)) + \sum_{j=1}^{n^{\text{Neg}}} exp(Sim(\mathbf{h}_i^1, \mathbf{h}_j^2))}. \tag{8}$$

Here, $Sim(\cdot)$ is the score function measuring the similarity between two nodes. For an anchor node $i$ in $G^1$, its representation is $\mathbf{h}_i^1$, we consider the nodes within functional module $\mathcal{M}_i$ from both graphs $G^1$ and $G^2$ as the positive samples, denoted as $\mathbf{h}_i^{pos}$, otherwise they are considered as negative samples. $n^{\text{Neg}}$ indicates the number of negative samples in a graph view. Consequently, the updated functional module-aware representation of node $i$ can be obtained, denoted as $\mathbf{h}_i$.

### 3.2.2 UPDATED SELF-ATTENTION MECHANISM

After obtaining the functional module-aware node representations, we design an updated self-attention mechanism. Inspired by Mialon et al. (2021), we design the self-attention mechanism as a kernel smoother to capture the similarity between each pair of nodes. Particularly, we define trainable exponential kernels on functional module-aware node representations. The updated self-attention is formulated as:

$$\text{FM-}Attn(i) = \sum_{j \in V} \frac{exp\Big(\langle \mathbf{W}_Q \mathbf{h}_i, \mathbf{W}_K \mathbf{h}_j \rangle / \sqrt{d_{\mathcal{K}}}\Big)}{\sum_{u \in V} exp\Big(\langle \mathbf{W}_Q \mathbf{h}_i, \mathbf{W}_K \mathbf{h}_u \rangle / \sqrt{d_{\mathcal{K}}}\Big)} f(\mathbf{h}_j). \tag{9}$$

Here, $exp\Big(\langle \mathbf{W}_Q \mathbf{h}_a, \mathbf{W}_K \mathbf{h}_b \rangle / \sqrt{d_{\mathcal{K}}}\Big)$ is a non-negative kernel, where $\langle \cdot, \cdot \rangle$ indicates the dot product. $f(\cdot)$ is a linear value function.

This updated self-attention mechanism can capture node similarity from the functional module level, without destroying the coherence of functional module-aware node representations. Representations of nodes in the same functional module are closer, while those from different modules keep farther. Therefore, the obtained node representations are more biologically plausible, preserving functional segregation and integration characteristics. The next theorem guarantees that our self-attention function can controllably preserve functional modules.

**Theorem 2** (Controllability analysis of functional module-aware self-attention). *Assume the functional module extractor $\psi$ is bounded by a constant $C_\psi$. For any two nodes $a$ and $b$, the distance between their representations after the functional module-aware self-attention is bounded by:*

$$\|\textit{FM-Attn}(a) - \textit{FM-Attn}(b)\| \leq C_\mathcal{M} \|\mathbf{h}_a - \mathbf{h}_b\|. \tag{10}$$

$\mathbf{h}_a := \psi(a, \mathcal{M}_a)$ *and* $\mathbf{h}_b := \psi(b, \mathcal{M}_b)$ *are representations of nodes $a$ and $b$ after the functional module extractor, respectively. $C_\mathcal{M}$ is a constant.*

This theorem demonstrates that node representations will maintain their relative distances after undergoing the functional module-aware self-attention mechanism. For example, after the self-attention, two nodes within the same functional module will remain close to each other, while two nodes from different functional modules will remain distant from each other. This is crucial for ensuring that the self-attention mechanism preserves functional modules while capturing the similarity between nodes. The proof of this theorem is provided in Appendix A.3.

## 4 EXPERIMENTS

### 4.1 EXPERIMENTAL SETUP

**Datasets.** We conduct experiments on fMRI data collected from three benchmark datasets. (1) Autism Brain Imaging Data Exchange (ABIDE) [3] dataset. This dataset contains resting-state fMRI data of $1,009$ anonymous subjects (age range: 5-64 years old) including $516$ Autism spectrum disorder patients and $493$ normal controls. The ROIs of brain graphs in ABIDE are defined by Craddock 200 atlas (Craddock et al., 2012). (2) Alzheimer's Disease Neuroimaging Initiative (ADNI) [4] dataset. The collected dataset comprises a total of $407$ subjects, including $190$ normal controls, $170$ mild cognitive impairment (MCI) patients, and $47$ Alzheimer's disease patients, carefully matched for both age and sex ratio. The ROI definition in ADNI dataset is based on AAL atlas (Tzourio-Mazoyer et al., 2002). (3) Attention Deficit Hyperactivity Disorder (ADHD-200) [5] dataset. This dataset contains $459$ subjects from 7 to 21 years old. $230$ subjects are typically developing individuals and $229$ subjects are ADHD patients. The ROI definition in ADHD-200 dataset is also based on Craddock 200 atlas. The number of ROIs in ABIDE, ADNI, and ADHD-200 datasets are 200, 90, and 190, respectively. Notably, brain graphs are constructed by computing the Pearson correlation coefficient (PCC) (Cohen et al., 2009) between ROIs based on the collected fMRI data. In particular, thresholds are set to keep edges with higher PCC values (weights) and drop those with lower PCC values (see Table 4 in Appdendix D.1).

**Evaluation Metrics.** We evaluate our model on the graph classification task. For ABIDE and ADHD-200 datasets, our model aims to detect whether the subject is a patient or a normal control. Therefore, the classification tasks in these two datasets are binary classification problems. For ADNI dataset, there are three groups, including normal controls, mild cognitive impairment patients, and Alzheimer's disease patients. Thus, disease detection in ADNI dataset is a multiple classification problem. We use five metrics to evaluate the model performance: (1) Test accuracy (ACC) indicates the ratio of brain graphs that are correctly classified out of all samples; (2) F1 score is the harmonic mean of precision and recall; (3) Area under the receiver operating characteristic curve (AUC) shows the trade-off between true positive rate and false positive rate; (4) Sensitivity (Sen.) refers to true positive rate; (5) Specificity (Spe.) gives the true negative rate. For the multiclass classification task on the ADNI dataset, we use macro averaging for the F1 score, sensitivity, and specificity. All results are the average values of $10$ random runs on test sets with the standard deviation.

---

[3] https://fcon_1000.projects.nitrc.org/indi/abide/
[4] https://adni.loni.usc.edu/
[5] https://fcon_1000.projects.nitrc.org/indi/adhd200/

**Baseline Methods.** We compare our model with state-of-the-art methods: (1) typical machine learning (ML) methods, including SVM (with a linear kernel) and Random Forest; (2) graph transformer models, including SAN (Kreuzer et al., 2021), Graph Transformer (Graph Trans.) (Dwivedi & Bresson, 2020), Graphormer (Ying et al., 2021), BRAINNETTF (Kan et al., 2022), SAT (Chen et al., 2022), Polynormer (Deng et al., 2024), Gradformer (Liu et al., 2024a), and GTSP (Liu et al., 2024b); (3) graph neural networks for brain graph analysis, including GAT (Velickovic et al., 2018), BrainGNN (Li et al., 2021), BrainGB (Cui et al., 2022), MCST-GCN (Zhu et al., 2024), and GroupBNA (Peng et al., 2024). For the SAT model, we consider its two variants as baselines: SAT without positional encoding (SAT-PE) and SAT with positional encoding (SAT+PE).

**Implementation Details.** Our model is implemented using PyTorch Geometric v2.0.4 and PyTorch v1.9.1. Model training is performed on an NVIDIA A6000 GPU with 48GB of memory. Our model is trained using the AdamW optimizer (Loshchilov & Hutter, 2019), and the cross-entropy loss is used for classification tasks. Each dataset is randomly split, with 80% used for training, 10% for validation, and 10% for testing. Full implementation is given in Appendix D.1.

## 4.2 RESULTS

The experimental results (ACC and AUC) on the three datasets are summarized in Table 1. The results for F1, Sen., and Spe. are provided in Appendix D.2. As experimental results show, the overall performance of BioBGT is superior to that of other baselines on all three datasets. For example, in the ADHD-200 dataset, we can see that the performance of BioBGT is distinguished, with the best accuracy and F1 score. Notably, our F1 score is around 4.21% higher than the second-best baseline. In the ABIDE dataset, BioBGT achieves a 5.76% improvement in accuracy over the second-best baseline. The experimental results demonstrate that our model excels in various brain disorder detection tasks.

Table 1: Results (mean ± margin of error) on three datasets (%).

| Method | | ADHD-200 | | ABIDE | | ADNI | |
|---|---|---|---|---|---|---|---|
| | | ACC | AUC | ACC | AUC | ACC | AUC |
| ML Methods | SVM | 53.56±2.73 | 54.66±3.40 | 49.01±1.70 | 49.05±1.94 | 32.29±2.63 | 49.88±3.10 |
| | Random Forest | 58.96±2.77 | 59.49±2.38 | 51.14±3.08 | 51.41±3.23 | 49.03±1.27 | 58.18±2.31 |
| Graph Transformer Models | SAN | 51.09±2.00 | 51.22±2.21 | 49.80±1.97 | 50.20±2.34 | 34.44±4.61 | 49.23±2.67 |
| | Graph Trans. | 50.76±2.07 | 51.49±1.15 | 50.20±0.50 | 48.20±0.16 | 40.28±4.17 | 52.31±2.04 |
| | Graphormer | 61.60±0.90 | 58.64 ±1.50 | 58.40±0.68 | 57.61±0.72 | 35.64±2.17 | 48.19±12.69 |
| | SAT-PE | 60.00±2.73 | 59.68±2.60 | 60.60±3.11 | 59.14±4.56 | 39.96±1.51 | 48.17±6.57 |
| | SAT+PE | 64.44±3.45 | 64.21±3.40 | 58.76±4.88 | 69.29±5.48 | 41.51±4.01 | 42.13±5.74 |
| | BRAINNETTF | 70.80±2.70 | **79.36±3.43** | 68.24±2.24 | **78.38±3.43** | 47.39±3.11 | 55.72±7.13 |
| | Polynormer | 64.78±2.34 | 63.61±2.43 | 57.03±0.96 | 56.42±1.56 | 41.85±2.12 | 54.34±4.37 |
| | Gradformer | 68.94±3.18 | 67.83±4.66 | 61.56±4.13 | 61.75±4.29 | 46.54±2.72 | 53.88±2.37 |
| | GTSP | 61.70±3.81 | 61.41±2.90 | 61.37±3.59 | 60.43±3.47 | 47.27±3.81 | 53.59±3.26 |
| Graph Neural Networks | GAT | 55.38±3.18 | 54.97±3.28 | 53.51±2.54 | 53.41±2.48 | 34.99±7.43 | 51.73±6.66 |
| | BrainGNN | 55.76±1.20 | 58.00±0.49 | 51.34±1.17 | 54.27±0.66 | 43.33±4.08 | 50.21±2.97 |
| | BrainGB | 68.20±7.81 | 74.64±10.10 | 65.12±3.90 | 70.32±3.66 | 44.34±3.90 | 62.24±4.68 |
| | MCST-GCN | 59.06±2.69 | 59.05±3.89 | 54.22±2.40 | 55.18±2.35 | 48.44±3.12 | 62.25±2.93 |
| | GroupBNA | 69.87±3.02 | 71.16±4.53 | 63.14±2.65 | 71.30±3.81 | 46.72±1.33 | 50.85±8.10 |
| Our Model | **BioBGT** | **71.06±0.08** | 71.64±1.14 | **74.00±2.01** | 73.33±2.37 | **52.08±2.08** | **62.33±5.98** |

## 4.3 ABLATION STUDIES

We conduct a series of ablation studies on three datasets to validate the effectiveness of each component in BioBGT. To verify how the network entanglement-based node importance encoding benefits the model performance, we conduct an ablation experiment by removing the node importance encoding, denoted as "-NE". In addition, to show the effectiveness of our functional module-aware self-attention, we remove the community contrastive strategy-based functional module extractor and replace our FM-$Attn(\cdot)$ with a normal self-attention $Attn(\cdot)$ (see Equation (2)), denoted as "-FM-$Attn$". Figure 3 compares the performance of BioBGT with the altered models on three datasets. BioBGT achieves superior performance compared to the model without node importance encoding

(-NE), indicating that our node importance encoding method is crucial for the model performance. Furthermore, BioBGT shows better performance than the altered model -FM-*Attn*. This indicates that it is essential to encode node similarities from the functional module level and preserve functional segregation and integration characteristics of brain graphs.

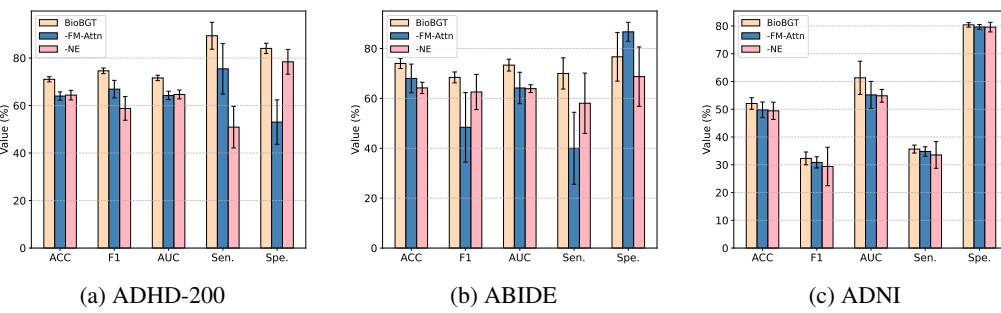

| (a) ADHD-200 | (b) ABIDE | (c) ADNI |

Figure 3: Model performance of BioBGT and its altered models.

## 4.4 COMPARATIVE ANALYSIS OF NODE IMPORTANCE MEASUREMENT

To validate the effectiveness of NE in measuring node importance, we perform a comparative analysis between BioBGT and its variants, which replace NE-based node importance encoding with (1) Laplacian matrix-based positional encoding, denoted as "+PE"; (2)degree centrality encoding, denoted as "+DC"; (3) Laplacian matrix and degree centrality encoding, denoted as "+PE+DC"; (4) betweenness centrality encoding, denoted as "+BC"; (5) closeness centrality encoding, denoted as "+CC"; (6) eigenvector centrality, denoted as "+EC". The results are summarized in Table 2 and Table 8 (see Appendix D.2). The overall performance of BioBGT is significantly better than other variants. This indicates that our node importance encoding method is crucial for the model performance, suggesting NE is more reliable for node importance measuring.

Table 2: The results (F1, ACC, AUC) for BioBGT and its variants on three datasets (%).

|  | ABIDE | | | ADNI | | | ADHD-200 | | |
|---|---|---|---|---|---|---|---|---|---|
|  | F1 | ACC | AUC | F1 | ACC | AUC | F1 | ACC | AUC |
| +PE | 54.00±2.97 | 60.60±2.25 | 60.91±2.05 | 30.09±3.36 | 49.43±2.42 | 52.14±2.41 | 69.21±7.14 | 67.56±3.24 | 67.39±2.80 |
| +DC | 59.73±4.23 | 61.20±1.88 | 61.28±1.80 | 27.27±1.57 | 50.57±1.64 | 55.45±3.59 | 74.22±1.35 | 69.78±2.33 | 70.18±2.28 |
| +PE+DC | 56.64±2.40 | 63.00±1.63 | 63.32±1.55 | 27.77±1.08 | 50.94±1.25 | 55.08±3.94 | 73.60±4.28 | 70.67±4.00 | 70.79±4.10 |
| +BC | 52.77±1.30 | 70.00±6.12 | 65.62±7.29 | 26.70±4.26 | 48.11±3.20 | 51.38±7.88 | 72.20±3.61 | 71.12±0.88 | 70.09±1,05 |
| +CC | 62.43±1.53 | 73.75±6.50 | 70.84±7.65 | 25.84±5.50 | 48.11±3.61 | 50.68±11.24 | 73.78±5.11 | **72.69±0.88** | 69.38±0.94 |
| +EC | 53.13±1.62 | 71.25±8.20 | 66.67±9.88 | 27.75±9.12 | 47.64±3.08 | 54.39±8.41 | 73.09±4.98 | 71.11±1.11 | 68.99±0.99 |
| **BioBGT** | **68.41±2.19** | **74.00±2.01** | **73.33±2.37** | **32.29±2.31** | **52.08±2.08** | **61.33±5.98** | **74.63±1.18** | 71.06±0.08 | **71.64±1.14** |

## 4.5 BIOLOGICAL PLAUSIBILITY ANALYSIS

We assess the biological plausibility of our node importance encodings and functional module-aware node representations by proving their consistency with existing neuroscience knowledge and providing reasonable explainability.

**Biological Plausibility in Node Importance Encoding.** Node efficiency (NEff) is one of the key metrics for complex graphs, which is used to measure the efficiency of information propagation between a given node and all other nodes in a graph (Latora & Marchiori, 2001). It can effectively quantify the communication strength of nodes, capturing their ability to facilitate information propagation across the graph. A high NEff value suggests that the node plays a crucial role as a hub for information propagation, enabling close and effective

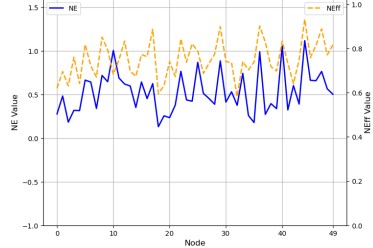

Figure 4: The NE and NEff values of 50 randomly selected nodes from a sample in the ABIDE dataset.

communication with the rest of the graph. Notably, NEff has been widely utilized in the field of neuroscience to identify critical regions in functional brain graphs (Bullmore & Sporns, 2009).

Therefore, we compare the NE values of each node with their corresponding NEff values to evaluate the validity of NE as a measure of node importance. Figure 4 presents the NE and NEff values of 50 randomly selected nodes from a sample randomly chosen in the ABIDE dataset. In addition, visualizations of NE and NEff values for all nodes in a randomly selected graph across three datasets are provided in Appendix D.3. As shown, the changing trend of the NE curve aligns closely with that of the NEff curve, demonstrating that nodes with high NE values also exhibit high NEff values, and vice versa. This consistency suggests that NE aligns with biological plausibility as a measure of node importance in information propagation. To further validate the effectiveness of NE in measuring node importance, we also compare each node's NE value to its average functional connectivity (FC) strength with all of the other nodes (see Appendix D.4).

**Biological Plausibility in Functional Module-Aware Self-Attention.** Figure 5 displays the heatmaps of the average self-attention scores from the ADHD-200 test set, output by Graphormer (a), SAT (b), and our functional module-aware self-attention mechanism (c). Based on the empirical labels of brain regions and functional modules (Dosenbach et al., 2007; 2010), the ROIs are classified into 6 functional modules, including visual cortex (Vis), motion control (MC), cognition control (CC), auditory cortex (Aud), language processing (LP), and executive control (EC). The detailed functional module division is provided in Appendix D.5. As illustrated in Figure 5, compared to the heatmaps produced by Graphormer and SAT, the heatmap of our model clearly shows that the learned attention scores of our model align better with the division of functional modules. For example, nodes within the visual cortex exhibit higher attention similarity. This observation verifies that our functional module-aware self-attention preserves the characteristics of functional segregation and integration within brain graph representations better. To further analyze the relationships between the attention patterns learned by our model and the biological patterns of target diseases, we visualize the self-attention scores of samples from each class in the three datasets (see Appendix D.6).

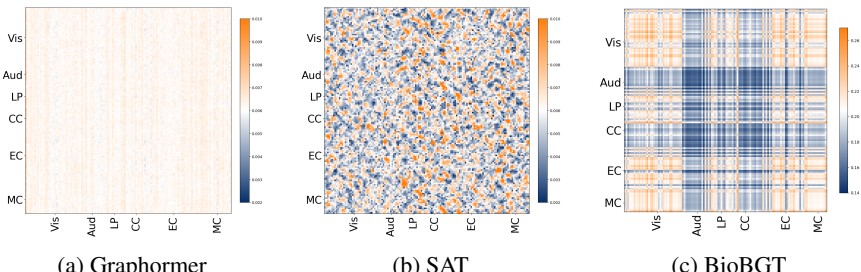

(a) Graphormer      (b) SAT      (c) BioBGT

Figure 5: The heatmaps of the average self-attention scores. Compared to other methods, heatmap (c) shows that learned attention scores of BioBGT align better with the division of functional modules.

## 5 RELATED WORK

### 5.1 BRAIN GRAPH ANALYSIS

Brain graphs, reflecting the connections in human neural system, are constructed from various brain health data, such as functional magnetic resonance imaging (fMRI), positron emission tomography (PET), and electroencephalography (EEG) (Bullmore & Sporns, 2009; Bessadok et al., 2022; Yu et al., 2024). Recently, graph learning-based brain graph analysis has attracted increased attention, dominating a range of tasks (e.g., brain disease detection and treatment recommendation) (Sun et al., 2024; Kan et al., 2022; Liu et al., 2023; Ding et al., 2023). NeuroGraph (Said et al., 2024) collects various brain connectome datasets for benchmarking graph learning models in brain graph analytical tasks (e.g., gender identification). BrainPrint (Wang et al., 2020b) develops a network estimation module and a graph analysis module to embed EEG features. BrainGNN (Li et al., 2021) contains special ROI-aware graph convolutional layers to capture the functional information of brain networks for fMRI analysis. BrainGB (Cui et al., 2022) summarizes the pipelines of brain graph construction. BRAINNETTF (Kan et al., 2022) utilizes a Transformer-based model to analyze brain graphs, while ignoring the structural encoding of ROIs and failing to preserve the small-world architecture of brain

graphs. MSE-GCN (Lei et al., 2023) applies multiple parallel graph convolutional network layers to encode brain structural and functional connectivities to detect early Alzheimer's disease (AD). GroupBNA (Peng et al., 2024) constructs group-specific brain networks via a group-adaptive brain network augmentation strategy.

## 5.2 GRAPH TRANSFORMERS

Graph transformers attempting to generalize Transformer models to graph data have shown significant performance in graph representation tasks (Kong et al., 2023; Luo et al., 2024b;a). SAN (Kreuzer et al., 2021) leverages the full Laplacian spectrum as the learned PE of input nodes, emphasizing the global structural information of the graph. Graphomer (Ying et al., 2021) introduces three SE methods to the Transformer architecture, including centrality encoding, spatial encoding and edge encoding, for graph representation learning. SAT (Chen et al., 2022) proposes a structure-aware self-attention mechanism to extract subgraph representations of nodes. SGFormer (Wu et al., 2024a) presents a single-layer attention model utilizing linear complexity to capture global dependencies among nodes. Geoformer (Wang et al., 2024) considers atomic environments as the PE of nodes in molecular graphs. EXPHORMER (Shirzad et al., 2023) proposes a sparse attention mechanism based on virtual global nodes and expander graphs for large graph representation learning. GRIT (Ma et al., 2023b) proposes a learned PE based on relative random walk probabilities and a flexible self-attention mechanism aiming to update both node and node-pair representations.

## 6 CONCLUSION

This paper presents the Biologically Plausible Brain Graph Transformer (BioBGT) model with a network entanglement-based node importance encoding technique and an updated functional module-aware self-attention mechanism. Extensive experiments on three benchmark datasets demonstrate our BioBGT outperforms state-of-the-art baselines, as well as enhances the biological plausibility of brain graph representations. Significantly, BioBGT offers valuable insights into enhancing the efficacy of brain graph analytical tasks, notably in the realm of improving disease detection. Our work could potentially advance digital health. Importantly, this work contributes to the intersection of neuroscience and artificial intelligence by proposing a brain graph representation learning technique that enhances biological plausibility.

While these findings are encouraging, some limitations remain. Firstly, according to current neuroscience knowledge, the biological properties of the brain are highly complex and remain uncertain, with many underlying mechanisms still requiring further research. Therefore, it is unlikely that a fully biologically plausible brain graph can be constructed. Instead, we can strive to build brain graphs that enhance biological plausibility by incorporating insights from existing knowledge, such as the brain's small-world architecture as one of its key features. Then, the computation complexity of network entanglement and the quadratic complexity of our functional module-aware self-attention module restrict the applicability of BioBGT. Although we have managed to keep the number of parameters comparable to other models, computational efficiency still needs future improvement. Therefore, it is worth exploring how to trade off the biological plausibility of brain graph representations and model computation complexity.

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

# A PROOFS

## A.1 PROOF OF PROPOSITION 1

**Proposition 1 (Density matrix as structural information.)** The structural information of a brain graph $G$, including the connection strength between nodes and the degree distribution of nodes, is encoded by its density matrix, which stands as a normalized information diffusion propagator and formulated as $\rho_G = \frac{e^{-\gamma \mathbf{L}}}{Z}$. Here, $e^{-\gamma \mathbf{L}}$ is the information diffusion propagator, $\gamma$ denotes the positive parameter, $\mathbf{L}$ is the Laplacian matrix of $G$, and $Z = \text{Tr}(e^{-\gamma \mathbf{L}})$ represents the partition function of $G$.

*Proof.* According to the theory of diffusion processes, an information diffusion process in a graph can be represented in exponential form as $e^{-\gamma \mathbf{L}}$ (Gasteiger et al., 2019; Li et al., 2024). Therefore, the density matrix can be regarded as a diffusion matrix along with structural information. In particular, the Laplacian matrix, $\mathbf{L} = \mathcal{D} - \mathcal{A}$, indicates the difference between the degree matrix $\mathcal{D}$ and the adjacency matrix $\mathcal{A}$. The degree matrix $\mathcal{D}$ is a diagonal matrix with its diagonal element $d_{ii}$ representing the degree of nodes $i$. The elements $a_{ij}$ of the adjacency matrix $\mathcal{A}$ represent the connection strengths between nodes $i$ and $j$. Therefore, the density matrix $\rho_G$ captures the connection strengths and degree distribution information among nodes in the graph $G$, encoding the global structural information of the graph $G$. $\qquad\square$

## A.2 PROOF OF THEOREM 1

**Theorem 1 (Quantification analysis of entanglement.)** Assume that the number of connected components in the $i$-control graph is the same as the original graph, denoted as $\alpha_i = \alpha$. The NE value of node $i$ is approximated as

$$\mathcal{NE}(i) \approx \left\| \frac{2m\gamma n^2}{\ln 2(n-\alpha)^2} \frac{\Delta Z}{Z Z_i} + \log_2\left(\frac{Z_i}{Z}\right) \right\|,$$

where, $n$ and $m$ are the numbers of nodes and edges, respectively. $Z_i$ stands as the partition function for $G_i$, and $\Delta Z = Z_i - Z$.

*Proof.* According to Equation (5) and spectral decomposition theory, the density matrix-based spectral entropy of graph $G$ is

$$\begin{aligned}
\mathcal{S}(G) &= \mathcal{S}(\rho_G) \\
&= -Tr(\rho_G \log_2 \rho_G) \\
&= -\sum_{j=1}^{n} \lambda_j(\rho_G) \log_2 \lambda_j(\rho_G),
\end{aligned}$$

where $\lambda_j(\rho_G) = \frac{e^{-\gamma \lambda_j(\mathbf{L})}}{Z}$. Therefore, we have

$$\begin{aligned}
\mathcal{S}(G) &= \frac{\gamma}{\ln 2} \sum_{j=1}^{n} \lambda_j(\mathbf{L}) \lambda_j(\rho_G) + \log_2 Z \\
&= \frac{\gamma}{\ln 2} \sum_{j=1}^{n} \langle \lambda_j(\mathbf{L}) \lambda_j(\rho_G) \rangle + \log_2 Z \\
&\approx \frac{\gamma}{\ln 2} \sum_{j=1}^{n} \langle \lambda_j(\mathbf{L}) \rangle \langle \lambda_j(\rho_G) \rangle + \log_2 Z.
\end{aligned}$$

The approximation comes from mean-field approximation. Assume the number of $\lambda_j$ satisfying $\lambda_j(\mathbf{L}) = 0$ is the same as the number of connected components $\alpha$. Refer to Huang et al. (2024), we

can get

$$\langle \lambda_j(\mathbf{L}) \rangle = \frac{1}{n} \sum_{j=1}^{n} \lambda_j(\mathbf{L})$$

$$= \frac{1}{n-\alpha} \sum_{j=\alpha+1}^{n} \lambda_j(\mathbf{L})$$

$$= \frac{2m}{n-\alpha}.$$

Likewise,

$$\langle \lambda_j(\rho_G) \rangle = \frac{1}{n} \sum_{j=1}^{n} \frac{e^{-\gamma \lambda_j(\mathbf{L})}}{Z}$$

$$= \frac{1}{n-\alpha} \sum_{j=\alpha+1}^{n} \frac{e^{-\gamma \lambda_j(\mathbf{L})}}{Z}$$

$$= \frac{1}{n-\alpha} (1 - \frac{\alpha}{Z}).$$

Therefore, $\mathcal{S}(G)$ can be approximated as

$$\mathcal{S}(G) \approx \frac{\gamma}{\ln 2} \frac{2mn}{n-\alpha} \cdot \frac{n}{n-\alpha} (1 - \frac{\alpha}{Z}) + \log_2 Z$$

$$= \frac{2m\gamma n^2}{\ln 2 (n-\alpha)^2} (1 - \frac{\alpha}{Z}) + \log_2 Z.$$

Then, we can quantify the perturbation of node $i$ on graph $G$ by capturing the changes of density matrix-based spectral entropy from $\mathcal{S}(G)$ to $\mathcal{S}(G_i)$, obtaining $\mathcal{NE}(i)$

$$\mathcal{NE}(i) = \|\mathcal{S}(G_i) - \mathcal{S}(G)\|$$

$$\approx \left\| \left( \frac{2m\gamma n^2}{\ln 2 (n-\alpha_i)^2} (1 - \frac{\alpha_i}{Z_i}) + \log_2 Z_i \right) - \left( \frac{2m\gamma n^2}{\ln 2 (n-\alpha)^2} (1 - \frac{\alpha}{Z}) + \log_2 Z \right) \right\|.$$

Suppose $\alpha_i = \alpha$, we can get

$$\mathcal{NE}(i) \approx \left\| \left( \frac{2m\gamma n^2}{\ln 2 (n-\alpha)^2} (1 - \frac{\alpha}{Z_i}) + \log_2 Z_i \right) - \left( \frac{2m\gamma n^2}{\ln 2 (n-\alpha)^2} (1 - \frac{\alpha}{Z}) + \log_2 Z \right) \right\|$$

$$= \left\| \frac{2m\gamma n^2}{\ln 2 (n-\alpha)^2} (\frac{1}{Z} - \frac{1}{Z_i}) + \log_2 (\frac{Z_i}{Z}) \right\|$$

$$= \left\| \frac{2m\gamma n^2}{\ln 2 (n-\alpha)^2} \frac{\Delta Z}{Z Z_i} + \log_2 (\frac{Z_i}{Z}) \right\|$$

Here, $\Delta Z = Z_i - Z$. $\qquad \square$

### A.3 PROOF OF THEOREM 2

**Theorem 2 (Controllability analysis of functional module-aware self-attention.)** Assume the functional module extractor $\psi$ is bounded by a constant $C_\psi$. For any two nodes $a$ and $b$, the distance between their representations after the functional module-aware self-attention is bounded by:

$$\|\text{FM-}Attn(a) - \text{FM-}Attn(b)\| \leq C_\mathcal{M} \|\mathbf{h}_a - \mathbf{h}_b\|.$$

$\mathbf{h}_a := \psi(a, \mathcal{M}_a)$ and $\mathbf{h}_b := \psi(b, \mathcal{M}_b)$ are representations of nodes $a$ and $b$ after functional module extractor, respectively. $C_\mathcal{M}$ is a constant.

*Proof.* The similarity representations of nodes $a$ and $b$ can be denoted as

$$\mathbf{z}_a = (\langle \mathbf{W}_Q \mathbf{h}_a, \mathbf{W}_K \mathbf{h}_i \rangle)_{i \in V} \in \mathbb{R}^n,$$
$$\mathbf{z}_b = (\langle \mathbf{W}_Q \mathbf{h}_b, \mathbf{W}_K \mathbf{h}_i \rangle)_{i \in V} \in \mathbb{R}^n.$$

Then, the $softmax(\mathbf{z})$ with the $k$-th coefficient is

$$softmax(\mathbf{z})_k = \frac{\exp(\mathbf{z}_k/\sqrt{d_\mathcal{K}})}{\sum_{j=1}^n \exp(\mathbf{z}_j/\sqrt{d_\mathcal{K}})}.$$

Afterwards, we can get

$$\|\text{FM-Attn}(a) - \text{FM-Attn}(b)\| = \left\| \sum_{i \in V} softmax(\mathbf{z}_a)_i f(\mathbf{h}_i) - \sum_{i \in V} softmax(\mathbf{z}_b)_i f(\mathbf{h}_i) \right\|$$

$$= \left\| \sum_{i \in V} \left( softmax(\mathbf{z}_a)_i - softmax(\mathbf{z}_b)_i \right) f(\mathbf{h}_i) \right\|$$

$$\leq \|softmax(\mathbf{z}_a) - softmax(\mathbf{z}_b)\| \sqrt{\sum_{i \in V} \|f(\mathbf{h}_i)\|^2}$$

$$\leq \frac{1}{\sqrt{d_\mathcal{K}}} \|\mathbf{z}_a - \mathbf{z}_b\| \sqrt{n} C.$$

The first inequality is based on a simple matrix norm inequality, and the second inequality is based on the fact that $softmax$ function is $\frac{1}{\sqrt{d_\mathcal{K}}}$-Lipschitz (Gao & Pavel, 2017). $C$ is the Lipschitz constant. Then, we can infer that

$$\|\mathbf{z}_a - \mathbf{z}_b\|^2 = \sum_{i \in V} \left( \langle \mathbf{W}_Q \mathbf{h}_a, \mathbf{W}_K \mathbf{h}_i \rangle - \langle \mathbf{W}_Q \mathbf{h}_b, \mathbf{W}_K \mathbf{h}_i \rangle \right)^2$$

$$= \sum_{i \in V} \left( \langle \mathbf{W}_Q (\mathbf{h}_a - \mathbf{h}_b), \mathbf{W}_K \mathbf{h}_i \rangle \right)^2$$

$$\leq \sum_{i \in V} \left( \|\mathbf{W}_Q (\mathbf{h}_a - \mathbf{h}_b)\|^2 \|\mathbf{W}_K \mathbf{h}_i\|^2 \right)$$

$$\leq \|\mathbf{W}_Q\|_\infty^2 \|\mathbf{h}_a - \mathbf{h}_b\|^2 C_\psi^2 \|\mathbf{W}_K\|_\infty^2$$

$$= C_\psi^2 \|\mathbf{W}_Q\|_\infty^2 \|\mathbf{W}_K\|_\infty^2 \|\mathbf{h}_a - \mathbf{h}_b\|^2.$$

The first inequality comes from the Cauchy-Schwarz inequality. The second inequality uses the definition of spectral norm, and constant $C_\psi$ is the bound of the functional module extractor $\psi$. Therefore, we can infer that

$$\|\text{FM-Attn}(a) - \text{FM-Attn}(b)\| \leq \frac{1}{\sqrt{d_\mathcal{K}}} \|\mathbf{z}_a - \mathbf{z}_b\| \sqrt{n} C.$$

$$\leq \sqrt{\frac{n}{d_\mathcal{K}}} C C_\psi \|\mathbf{W}_Q\|_\infty \|\mathbf{W}_K\|_\infty \|\mathbf{h}_a - \mathbf{h}_b\|.$$

A constant $C_\mathcal{M}$ can be defined as

$$C_\mathcal{M} = \sqrt{\frac{n}{d_\mathcal{K}}} C C_\psi \|\mathbf{W}_Q\|_\infty \|\mathbf{W}_K\|_\infty.$$

Consequently, we can obtain the inequality in the Theorem 2.

$$\|\text{FM-}Attn(a) - \text{FM-}Attn(b)\| \leq C_\mathcal{M} \|\mathbf{h}_a - \mathbf{h}_b\|.$$

$\square$

## B    RELIABILITY OF NE FOR NODE IMPORTANCE MEASURING

Given a node $i$, its importance degree in the information propagation across the graph is defined as its NE value, which is obtained by measuring the disparity between the density matrix-based spectral entropy of the original graph and that of the $i$-control graph. The density matrix-based spectral entropy

captures both the global topology and information diffusion process of the graph (Huang et al., 2024). Therefore, NE can fully reflect the significance of a node on the information propagation across the graph. For example, if the density matrix-based spectral entropy of the $i$-control graph changes greatly compared to the original graph, it indicates that induced perturbation of node $i$ can lead to significant changes in the global topology and information diffusion pattern of the graph. This means node $i$ plays an important role in the graph in terms of information propagation. Particularly, brain graphs are communication networks, in which information propagation is a crucial aspect (Seguin et al., 2023). Hence, it is essential to measure node importance based on NE reflecting on the changes in the global topology and information diffusion of brain graphs.

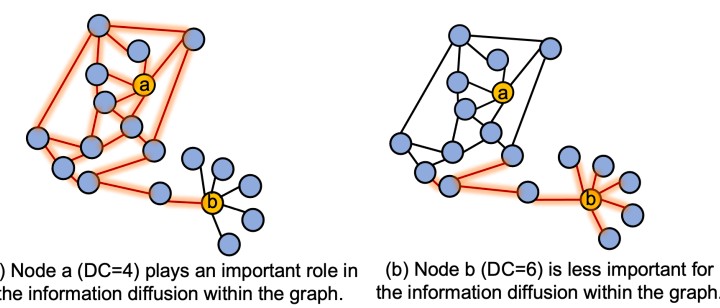

(a) Node a (DC=4) plays an important role in the information diffusion within the graph.

(b) Node b (DC=6) is less important for the information diffusion within the graph.

Figure 6: An example of the fragility of DC in measuring node importance in brain graphs. Highlighted edges represent paths of information diffusion.

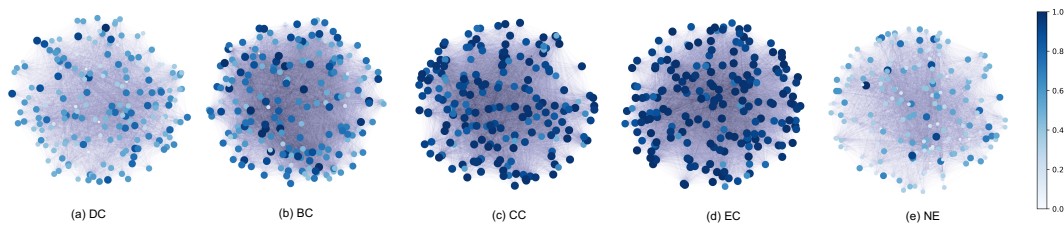

(a) DC    (b) BC    (c) CC    (d) EC    (e) NE

Figure 7: Visualization of the DC (a), BC (b), CC (c), EC (d), and NE (e) values of nodes in a brain graph from the ABIDE dataset. The colors of nodes are proportional to the normalized values of the DC and NE.

Due to the capability of NE to reflect the global topology and information diffusion, it is more reliable for node importance measuring in brain graphs compared to other methods, such as centralities (Das et al., 2018). Four representative centralities, including degree centrality (DC), closeness centrality (CC), betweenness centrality (BC), and eigenvector centrality (EC), emphasize the local structure or local message passing, having limitations to fully capture both global topology and information diffusion process. Therefore, they are not reliable for measuring node importance, especially for brain graphs. For example, Figure 6 shows the fragility of DC in measuring node importance in brain graphs. Node $a$ plays a more important role in information propagation than node $b$, while its DC value (=4) is lower than that of node $b$ (=6). We also visualize the DC, BC, CC, EC, and NE values of nodes in a real brain graph from the ABIDE dataset [6] (see Figure 7). As shown in Figure 7, NE can better distinguish node importance in the entire brain graph. In contrast, other methods may mistakenly identify marginal nodes with many direct connections but less significance in information propagation as important nodes. For example, especially in Figure 7 (c) and (d), many nodes exhibit disproportionately high CC and EC values, highlighting their inability to effectively differentiate between truly important and less significant nodes in information propagation.

---

[6]https://fcon_1000.projects.nitrc.org/indi/abide/

## C    EDGE DROPPING STRATEGY

Following  Chen et al. (2023), we apply the edge dropping strategy to achieve graph augmentation. The main idea of our edge dropping strategy is dropping less important edges while preserving the functional module structure. Particularly, our edge dropping strategy is based on inter-modular edges first rationale. That is inter-modular edges are less important than intra-module edges. We define a scoring function $IM(\cdot)$ to calculate the importance of each edge, it must meet the following condition:

$$IM(e_{intra}) > IM(e_{inter}). \tag{11}$$

Here, $e_{intra}$ and $e_{inter}$ are intra-module and inter-modular edges, respectively. If nodes $i$ and $j$ are from the same functional module, the scoring function of their edge $e_{ij}$ is:

$$IM(e_{ij}) = \omega_{e_{ij}} + max(\omega), \tag{12}$$

where $\omega_{e_{ij}}$ is the weight of edge $e_{ij}$, and $max(\omega)$ indicates the highest edge weight in the graph. On the other hand, if nodes $i$ and $j$ are from different functional modules, the scoring function of their edge $e_{ij}$ is:

$$IM(e_{ij}) = \omega_{e_{ij}} - max(\omega). \tag{13}$$

Consequently, the scoring function of inter-module edges will keep lower than that of intra-module edges.  Our edge dropping strategy will first consider dropping inter-module edges with lower importance scores. Therefore, the functional module structure can be preserved in two augmented graphs.

## D    EXPERIMENTAL DETAILS AND ADDITIONAL RESULTS

### D.1    IMPLEMENTATION DETAILS

The detailed hyperparameter settings for training BioBGT on three datasets are summarized in Table 3.

Table 3: Hyperparameters for training BioBGT on three different datasets.

| Hyperparameter | ABIDE | ADNI | ADHD-200 |
|---|---|---|---|
| #Layers | 3 | 3 | 3 |
| #Attention heads | 8 | 8 | 8 |
| Threshold of edge weight | 0.3 | 0 | 0 |
| Hidden dimensions | 128 | 128 | 128 |
| FFN hidden dimensions | 256 | 256 | 256 |
| Dropout | 0.5 | 0.1 | 0.1 |
| Readout method | mean | mean | mean |
| Learning rate | 3e-4 | 3e-4 | 3e-4 |
| Batch size | 128 | 128 | 128 |
| #Epochs | 200 | 200 | 200 |
| Weight decay | 1e-4 | 1e-4 | 1e-4 |
| Warm-up Steps | 10 | 10 | 10 |

Table 4: The number of parameters for different models on three datasets.

| Method | ABIDE | ADNI | ADHD-200 |
|---|---|---|---|
| Polynormer | 9.78M | 9.80M | 9.83M |
| Random Forest | 21K | 18K | 9K |
| Gradformer | 455K | 446K | 468K |
| GTSP | 427K | 424K | 430K |
| BioBGT | 465K | 455K | 454K |

**Number of parameters and computation time.** The number of parameters for BioBGT are 465K, 455K, and 454K on ABIDE, ADNI, and ADHD-200, respectively. The running time for training BioBGT on ABIDE, ADNI, and ADHD-200 is 650 s/epoch, 265 s/epoch, and 251 s/epoch, respectively. Table 4 presents the number of parameters for different models on three datasets.

## D.2 Additional Experimental Results

Table 5, Table 6, and Table 7 give the additional results (F1, Sen., Spe.) for BioBGT compared to state-of-the-art methods on ADHD-200, ABIDE, and ADNI datasets, respectively. The overall experimental result reveals that BioBGT outperforms other methods, suggesting its superiority in various brain graph analysis tasks. Please note that due to the randomness of experimental results, the reproduced results of some baselines may be a little different from those in the original papers. Table 8 summarizes the results of sensitivity and specificity for BioBGT and its variants on three datasets.

Table 5: Additional results on ADHD-200 (%).

| Method | | F1 | Sen. | Spe. |
|---|---|---|---|---|
| ML Methods | SVM | 58.91±5.47 | 68.75±16.11 | 40.58±18.58 |
| | Random Forest | 57.07±4.13 | 55.01±9.30 | 63.95±8.04 |
| Graph Transformer Models | SAN | 48.19±9.76 | 50.43±19.32 | 51.74±20.16 |
| | Graph Trans. | 55.58±4.18 | 62.39±9.43 | 39.13±10.74 |
| | Graphormer | 69.25±3.05 | 83.34±2.90 | 33.96±6.10 |
| | SAT-PE | 65.37±1.61 | 73.91±3.73 | 45.45±13.20 |
| | SAT+PE | 68.30±3.83 | 75.65±7.40 | 52.73±10.67 |
| | BRAINNETTF | 70.42±3.06 | 66.69±4.93 | 75.75±4.63 |
| | Polynormer | 58.62±12.93 | 51.57±17.68 | 69.63±11.59 |
| | Gradformer | 69.01±5.39 | 71.66±9.37 | 63.99±12.97 |
| | GTSP | 64.59±7.29 | 75.60±10.47 | 47.22±14.84 |
| Graph Neural Networks | GAT | 57.85±5.87 | 64.54±13.97 | 45.10±18.34 |
| | BrainGNN | 55.89±1.21 | 56.09±2.13 | 55.43±3.11 |
| | BrainGB | 63.73±11.93 | 58.26±15.82 | 78.20±8.19 |
| | MCST-GCN | 54.76±1.20 | 30.42±2.17 | 68.05±1.79 |
| | GroupBNA | 70.61±2.35 | 74.05±4.60 | 72.93±6.49 |
| Our Model | **BioBGT** | **74.63±1.18** | **89.39±5.66** | **84.07±2.19** |

Table 6: Additional results on ABIDE (%).

| Method | | F1 | Sen. | Spe. |
|---|---|---|---|---|
| ML Methods | SVM | 52.02±2.16 | 54.41±5.44 | 43.69±1.90 |
| | Random Forest | 51.46±3.87 | 51.23±7.82 | 51.58±5.75 |
| Graph Transformer Models | SAN | 47.52±3.74 | 44.95±6.53 | 54.85±7.48 |
| | Graph Trans. | 57.98±0.52 | 67.38±0.78 | 32.32±0.45 |
| | Graphormer | 64.17±2.89 | **70.64±8.74** | 46.49±12.51 |
| | SAT-PE | 67.21±3.84 | 64.76±8.62 | 53.52±15.34 |
| | SAT+PE | 68.33±2.88 | 69.29±5.48 | 43.24±18.03 |
| | BRAINNETTF | 68.20±2.31 | 69.39±5.22 | 66.95±5.28 |
| | Polynormer | 44.20±11.13 | 38.16±17.04 | 74.68±15.46 |
| | Gradformer | 62.12±3.37 | 59.16±4.77 | 64.35±10.99 |
| | GTSP | 65.36±6.22 | 69.86±11.57 | 50.99±12.59 |
| Graph Neural Networks | GAT | 56.12±6.25 | 57.73±12.96 | 48.13±12.44 |
| | BrainGNN | 50.09±1.55 | 54.85±4.12 | 47.96±2.92 |
| | BrainGB | 66.95±5.08 | 67.01±10.00 | 60.07±8.53 |
| | MCST-GCN | 52.63±6.33 | 48.80±6.88 | 45.85±10.19 |
| | GroupBNA | 61.64±1.20 | 64.72±5.43 | 65.28±6.27 |
| Our Model | **BioBGT** | **68.41±2.19** | 70.00±6.25 | **76.67±9.77** |

## D.3 Additional NE and NEff Curves for Three Datasets

We provide visualizations of NE and NEff values for all nodes in brain graphs across three datasets. The graphs from the ABIDE, ADHD-200, and ADNI datasets contain 200, 190, and 90 nodes,

Table 7: Additional results on ADNI (%).

| Method | | F1 | Sen. | Spe. |
|---|---|---|---|---|
| ML Methods | SVM | 26.69±4.56 | 27.33±5.12 | 74.91±0.63 |
| | Random Forest | 29.18±2.13 | 32.33±1.11 | 79.03±0.34 |
| Graph Transformer Models | SAN | 14.73±3.01 | 23.24±2.65 | 74.30±1.41 |
| | Graph Trans. | 20.96±1.51 | 26.65±2.24 | 76.14±1.22 |
| | Graphormer | 21.63±5.85 | 25.72±4.97 | 75.78±3.47 |
| | SAT-PE | 24.08±4.18 | 30.67±4.19 | 76.16±0.29 |
| | SAT+PE | 19.66±5.13 | 26.72±3.46 | 76.16±2.15 |
| | BRAINNETTF | 35.64±3.97 | 35.25±3.29 | 80.03±4.43 |
| | Polynormer | 16.75±3.05 | 25.72±0.92 | 75.47±0.60 |
| | Gradformer | 23.80±2.61 | 29.90±1.92 | 78.18±1.24 |
| | GTSP | 25.48±4.14 | 30.39±1.91 | 78.26±1.00 |
| Graph Neural Networks | GAT | 25.70±3.54 | 30.92±4.01 | 77.72±1.40 |
| | BrainGNN | 23.23±4.67 | 29.52±3.31 | 77.47±1.80 |
| | BrainGB | 32.22±7.96 | 34.04±6.48 | 78.78±1.54 |
| | MCST-GCN | **37.44±3.12** | **38.06±2.99** | 70.52±2.71 |
| | GroupBNA | 35.85±1.38 | 34.98±9.72 | 78.25±6.78 |
| Our Model | **BioBGT** | 32.29±2.31 | 35.65±1.43 | **80.39±0.82** |

Table 8: The results (Sen. and Spe.) for BioBGT and its variants on three datasets (%).

| | ABIDE | | ADNI | | ADHD-200 | |
|---|---|---|---|---|---|---|
| | Sen. | Spe. | Sen. | Spe | Sen. | Spe |
| +PE | 45.49±4.36 | 76.33±7.05 | 34.50±2.36 | 79.44±1.15 | 74.78±18.66 | 60.00±12.80 |
| +DC | 57.26±9.11 | 65.31±9.32 | 32.68±1.05 | 79.69±0.73 | 88.42±9.08 | 51.94±12.25 |
| +PE+DC | 47.45±2.84 | 79.18±2.28 | 32.99±0.88 | 79.82±0.73 | 82.53±12.77 | 59.05±10.57 |
| +BC | 43.75±13.98 | 87.50±4.17 | 31.53±2.66 | 78.71±1.36 | 77.12±13.78 | 74.75±7.46 |
| +CC | 56.25±11.92 | 85.41±3.61 | 31.50±3.20 | 78.84±1.80 | 77.17±15.56 | 71.59±9.74 |
| +EC | 43.75±8.30 | **89.58±3.61** | 33.99±7.13 | 78.99±1.27 | 81.52±18.57 | 70.46±8.37 |
| **BioBGT** | **70.00±6.25** | 76.67±9.77 | **35.65±1.43** | **80.39±0.82** | **89.39±5.66** | **84.07±2.19** |

respectively. Figure 8 provides the curves for all 200 nodes from a randomly selected sample in the ABIDE dataset. Figure 9 shows the curves for all 190 nodes from a randomly selected sample in the ADHD-200 dataset. Figure 10 shows the curves for all 90 nodes from a randomly selected sample in the ADNI dataset.

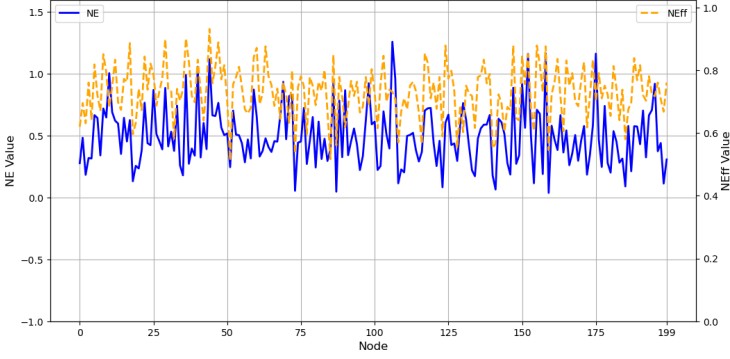

Figure 8: The NE and NEff values of 200 nodes from a randomly selected sample in the ABIDE dataset.

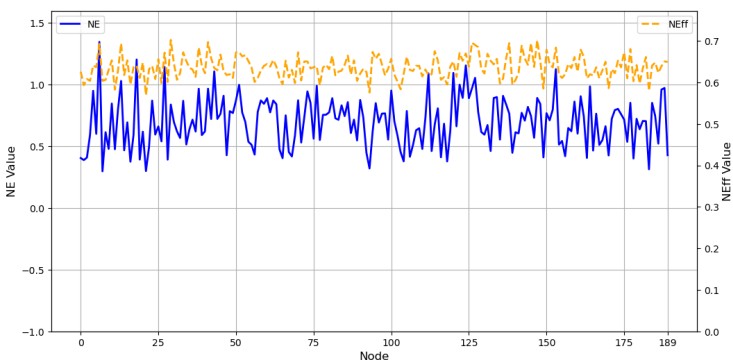

Figure 9: The NE and NEff values of 190 nodes from a randomly selected sample in the ADHD-200 dataset.

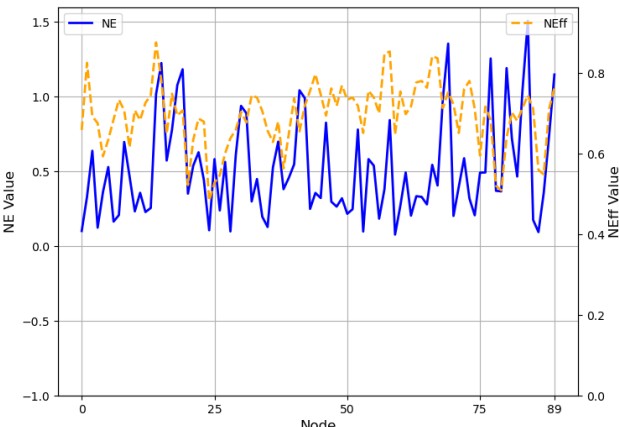

Figure 10: The NE and NEff values of 90 nodes from a randomly selected sample in the ADNI dataset.

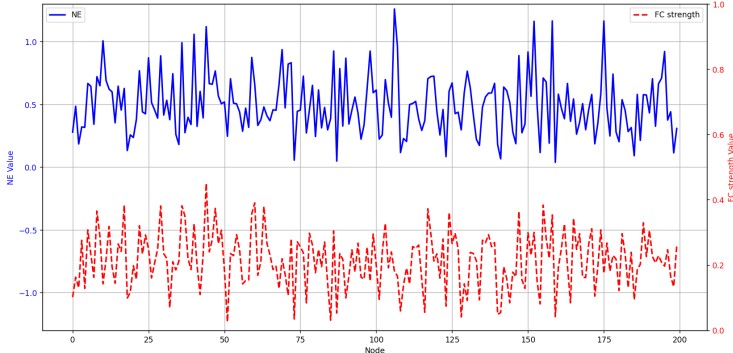

Figure 11: The NE value and FC strength of 200 nodes from a randomly selected sample in the ABIDE dataset.

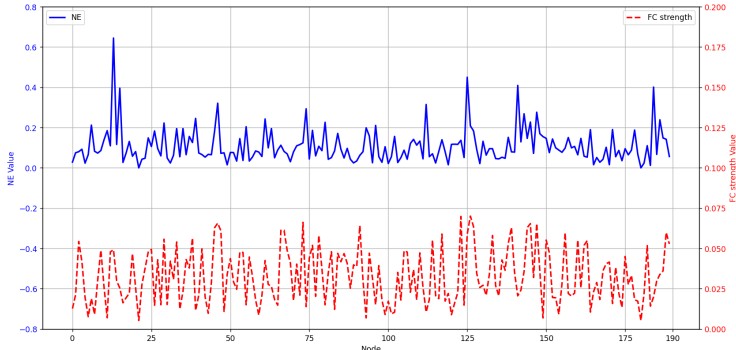

Figure 12: The NE value and FC strength of 190 nodes from a randomly selected sample in the ADHD-200 dataset.

## D.4 NE AND FC STRENGTH CURVES FOR THREE DATASETS

To further validate the effectiveness of NE in measuring node importance, we compare each node's NE value to its average functional connectivity (FC) strength with all of the other nodes. FC strength is widely regarded as a suitable metric for capturing the global topological properties of functional brain graphs (Liang et al., 2012). Notably, FC strength is commonly quantified using the Pearson correlation coefficient (PCC) between nodes, a standard measure in brain graph analysis (Li et al., 2021). A node's average FC strength (average PCC value) reflects its communication strength with other nodes, providing an additional perspective on its importance in information propagation. Figures 11, 12, and 13 illustrate the NE and FC strength curves for all nodes in randomly selected samples from the ABIDE, ADHD-200, and ADNI datasets, respectively. The results reveal an obvious alignment between the trends of the NE and FC strength curves, indicating that nodes with high NE values also exhibit high FC strength, and vice versa. This consistency further supports the validity of NE as a measure for quantifying node importance in information propagation.

## D.5 FUNCTIONAL MODULE DIVISION

Table 9 gives the empirical labels of ROIs and functional modules. The functional modules used in this study are based on empirical labels. These labels represent the best effort to categorize regions based on known functional associations, but they are inherently limited due to the complex biological properties of the brain graph. Empirical functional modules often encompass ROIs from diverse brain regions, resulting in heterogeneity within each module. For example, in the auditory cortex, both temporal lobe regions (e.g., 'temporal 103') and thalamic regions (e.g., 'thalamus 57') are included due to their involvement in auditory processing (Rauschecker & Scott, 2009; Jones, 2012). This diversity may reduce the uniformity of high self-attention scores within the module. In addition,

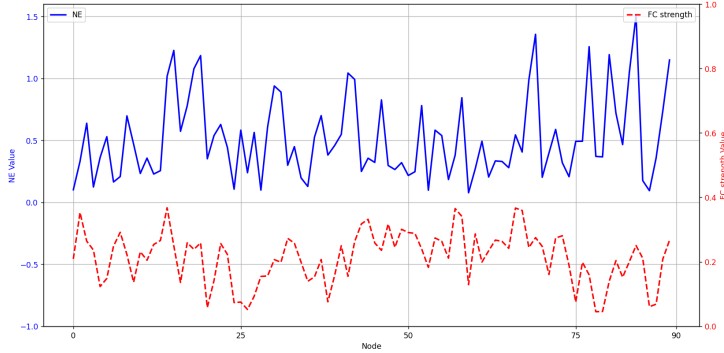

Figure 13: The NE value and FC strength of 90 nodes from a randomly selected sample in the ADNI dataset.

Table 9: The labels of ROIs and functional modules.

| Functional Module | ROI |
|---|---|
| Vis | 'occipital 146', 'occipital 141', 'occipital 136', 'occipital 137', 'occipital 92', 'occipital 149', 'occipital 148', 'occipital 145', 'occipital 147', 'occipital 142', 'occipital 139', 'occipital 135', 'occipital 133', 'occipital 129', 'occipital 126', 'occipital 118', 'occipital 119', 'occipital 106', 'post occipital 160', 'post occipital 158', 'post occipital 159', 'post occipital 157', 'post occipital 156', 'post occipital 153', 'post occipital 154', 'post occipital 152' |
| MC | 'inf cerebellum 155', 'inf cerebellum 150', 'inf cerebellum 151', 'inf cerebellum 140', 'inf cerebellum 131', 'inf cerebellum 122', 'inf cerebellum 121', 'inf cerebellum 110', 'lat cerebellum 128', 'lat cerebellum 113', 'lat cerebellum 109', 'lat cerebellum 98', 'med cerebellum 143', 'med cerebellum 144', 'med cerebellum 138', 'med cerebellum 130', 'med cerebellum 127', 'med cerebellum 120', 'post parietal 99', 'SMA 43', 'basal ganglia 71', 'basal ganglia 38', 'basal ganglia 39', 'basal ganglia 30' |
| CC | 'post cingulate 115', 'post cingulate 111', 'post cingulate 108', 'post cingulate 93', 'post cingulate 90', 'post cingulate 73', 'vlPFC 15', 'vmPFC 13', 'vmPFC 11', 'vmPFC 7', 'vmPFC 6', 'vmPFC 1', 'IPL 107', 'IPL 104', 'IPL 101', 'IPL 96', 'IPL 88', 'IPS 116', 'IPS 114' |
| Aud | 'sup temporal 100', 'temporal 103', 'temporal 95', 'temporal 78', 'thalamus 57', 'thalamus 58', 'thalamus 47', 'inf temporal 91', 'inf temporal 72', 'inf temporal 63', 'temporal 123' |
| LP | 'angular gyrus 102', 'mid insula 61', 'mid insula 59', 'mid insula 44', 'aPFC 5', 'angular gyrus 124', 'angular gyrus 117', 'aPFC 2', 'aPFC 3' |
| EC | 'dACC 27', 'dFC 36', 'dFC 34', 'dFC 29', 'dlPFC 24', 'dlPFC 22', 'vPFC 23', 'vent aPFC 10', 'vent aPFC 9','vlPFC 12','dlPFC 16','dFC 3' |

because of limited label availability, there are no available labels for the atlas of the ADNI and ABIDE datasets. Therefore, we can only provide labels for the ADHD-200 dataset.

D.6 ANALYSIS OF ATTENTION PATTERNS OUTPUT BY BIOBGT

**Comparsion between NCs and ADHD patients.** Studies in neuroscience have shown that functional connectivity (FC) in ADHD patients can be either enhanced or weakened to varying degrees, disrupting functional modularity (Wang et al., 2020a). For instance, Sripada et al. demonstrated that the FC strength within the cognition control (CC) module is reduced in individuals with ADHD (Sripada et al., 2014). Figure 14 presents the heatmaps of the average self-attention scores for normal controls (NCs) and ADHD patients from the ADHD-200 test set. It is evident that the functional modules in the heatmap of ADHD patients are less distinct, while the NC group's heatmap shows clearer functional modules, such as Vis, CC, and MC modules. Specifically, the CC module in the NC group exhibits higher internal strength and stronger modularity compared to the CC module in the ADHD group, which aligns with Sripada et al.'s findings in neuroscience.

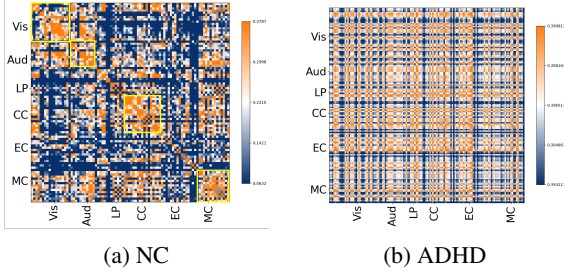

(a) NC          (b) ADHD

Figure 14: The heatmaps of the average self-attention scores of normal controls (a) and ADHD patients (b) in the ADHD-200 test set.

**Comparsion between NCs and ASD patients.** Studies have shown that the small-world properties of brain graphs in autism spectrum disorder (ASD) patients are generally lower than those of typically developing individuals (Itahashi et al., 2014). Particularly, the hub properties of nodes are weakened in brain graphs of ASD patients. Figure 15 shows the heatmaps of the average self-attention scores for NCs and ASD patients from the ABIDE test set. It can be observed that the attention heatmap of NCs contains nodes with strong hub properties, indicating high correlations with most other nodes. In contrast, the attention heatmap of ASD patients shows weaker hub properties, which aligns with the finding that the small-world properties of functional connectivity in ASD patients are diminished.

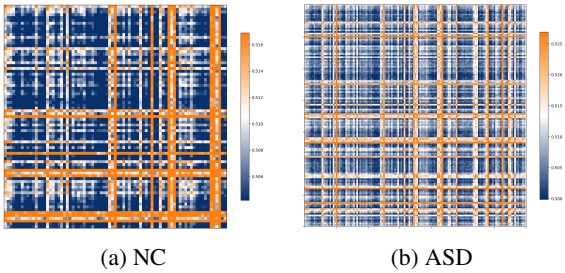

(a) NC          (b) ASD

Figure 15: The heatmaps of the average self-attention scores of normal controls (a) and ASD patients (b) in the ABIDE test set.

**Comparsion between NCs, MCI patients, and AD patients.** Different stages of Alzheimer's disease exhibit distinct functional connectivity patterns (Sanz-Arigita et al., 2010). Specifically, as the disease progresses, the modular characteristics of functional brain graphs decline (Dai & He, 2014; Zhao et al., 2012). The brain graphs of NCs display more distinct modular structures, while the functional connectivity patterns of mild cognitive impairment (MCI) patients begin to show signs of disorganization. For patients with Alzheimer's disease (AD), the partitioning of functional modules almost disappears, and modularity is significantly diminished. Figure 16 presents the heatmaps of the average self-attention scores for NCs (a), MCI patients (b), and AD patients (c) in the ADNI test set. It is evident that the brain graphs of NCs exhibit more distinct functional modules compared to those of MCI and AD patients.

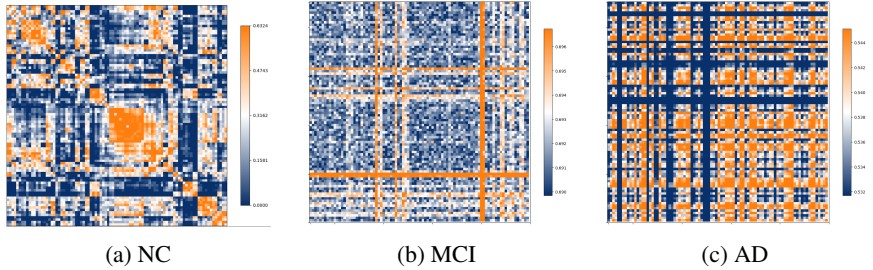

(a) NC          (b) MCI          (c) AD

Figure 16: The heatmaps of the average self-attention scores normal controls (a), MCI patients (b), and AD patients (c) in the ADNI test set.

Table 10: Model performance on Citeseer and Cora (%).

|  | Citeseer | | | Cora | | |
|---|---|---|---|---|---|---|
|  | F1 | ACC | AUC | F1 | ACC | AUC |
| GAT | 45.45±5.17 | **70.01±0.68** | 81.06±1.91 | 81.02±0.46 | 81.74±0.53 | **97.38±0.03** |
| SAT+PE | 64.04±1.25 | 68.80±0.52 | 87.69±1.99 | 81.62±0.56 | 82.84±0.87 | 94.63±0.34 |
| **BioBGT** | **64.80±0.34** | 69.02±0.70 | **88.04±1.51** | **82.03±1.33** | **83.14±0.80** | 94.58±0.24 |

### D.7 MODEL GENERALIZABILITY ANALYSIS

We suggest that BioBGT not only performs well on brain graphs but may also generalize effectively to other networks with similar structural characteristics, such as the presence of hubs and modules. To prove this, we apply BioBGT to other types of networks, such as citation networks. We train our model on the Citeseer (Giles et al., 1998) and Cora (McCallum et al., 2000) datasets for node classification tasks. Table 10 highlights the superiority of BioBGT on both Citeseer and Cora datasets. Therefore, our model shows generalizability in extending to other networks that contain hubs and modules.

