# OpenReview forum: "Biologically Plausible Brain Graph Transformer"
_ICLR.cc/2025/Conference — ICLR 2025 Poster_

### Official Review · Reviewer_drzD · 2024-10-31

**Soundness:** 3
**Presentation:** 4
**Contribution:** 4
**Rating:** 8
**Confidence:** 4

**Summary:**

The Biologically Plausible Brain Graph Transformer (BioBGT) is introduced here as a graph transformer specifically aimed at creating brain graph representations rooted in biological principles. Unlike prior models that allow representations to form without these constraints, BioBGT uses the concept of network entanglement - borrowed from quantum computing - to more accurately assess node importance than traditional centrality metrics. Furthermore, the authors implement a community contrastive training method, enabling node representations to account for network communities. These two contributions are aligned with core characteristics of fMRI-based graphs, underscoring their relevance for neuroimaging-focused models.


_____________________
EDIT (after rebuttal period): I appreciate the answers and discussions that the authors had with me and the other reviewers. Based on all this I'm increasing the score from 6 to 8 (accept), as well as increasing the contribution score from 3 to 4. There are still some small weaknesses that I've mentioned in this discussion and that is why I do not increase the score of this paper to the maximum value. I thank the authors for the insightful discussions and the time taken to address my concerns.

**Strengths:**

I found this paper compelling and I believe it has strong potential for ICLR. The writing is clear and well-organized, making it highly readable. The paper’s originality lies in its combination of concepts from another field, applied thoughtfully to the fMRI domain, with evidence that this approach yields positive outcomes. In this sense, the results in tables 1 and 2 are quite impressive, while those in Table 6, though somewhat less robust, still demonstrate solid performance.

The authors leverage three widely recognized neuroimaging datasets and employ two different parcellation methods. I feel the choice of baseline models is appropriate for the study’s goals, with an exception that I will provide in the next section. The quick experiments in Section 4.5 are noteworthy, showing that the model can achieve comparable performance on non-neuroimaging datasets with similar underlying motivations.

**Weaknesses:**

1. No experiment is conducted with a smaller parcellation. The results using the ADNI dataset - where a smaller parcellation was used - are not as strong, which in my opinion are an indication that a smaller number of ROIs could impact the performance of this method. Given the focus on community structure and node importance of this work, I think it would have been important to understand whether the size of the parcellations are an important factor in this models performance.
2. Even though the ablation study makes a good job in comparing different alternatives to the node importance encoding by comparing other centrality measures, there is no ablation study for the community contrastive strategy.
3. The work lacks comparison with more "traditional" ML models like random forests or SVM. In a few fMRI works/papers that I have seen in the past, and also based on my experience with these datasets, I typically see is that more "traditional" models achieve similar or even better performance than deep learning when trained on the upper-triangle of the correlation matrices (which has been historically widely explored in the connectomics field). In this sense, for a conference like ICLR, I find the lack of comparisons with traditional ML models in this specific context a weakness of this paper.
4. This work has an entire section (4.4) focused on the biological plausibility of its model, but these results do not discuss whether these modules correspond to any known brain structure or medical knowledge. Furthermore, it is not clear how these different modules contribute to the final classification, and thus it is not clear how plausible and good these representations are.

**Questions:**

I will be happy to revise my final score if the authors tackle the weaknesses I have identified. Furthermore, I have the following questions, more or less in the order they appear in the paper:
1. Equation 7 mentions "learnable embedding vectors", but I do not see those in the equation. Can the authors specify this better?
2. Did the authors try other contrastive loss functions beyond the one in equation 8?
3. How did the authors calculate the F1/sensitivity/specificity scores for the ADNI dataset, where we have 3 classification labels?
4. In Conclusion, the authors argue that they "managed to keep the number of parameters comparable to other models". Where is this shown in the paper?

---

> ### Author Response · Authors · 2024-11-20
> **Response to Reviewer drzD (Part 1)**
>
> Thank you very much for pointing out the issues in our paper and providing your insightful suggestions. Based on your comments, we have revised the manuscript (green text in the manuscript) and provided detailed responses below.
>
> **Weakness 1**: No experiment is conducted with a smaller parcellation. The results using the ADNI dataset - where a smaller parcellation was used - are not as strong, which in my opinion are an indication that a smaller number of ROIs could impact the performance of this method. Given the focus on community structure and node importance of this work, I think it would have been important to understand whether the size of the parcellations are an important factor in this models performance.
>
> **Response**: We agree that the size of the parcellations is indeed an important factor in model performance. We apologize for not clearly stating the number of nodes in each dataset. In our experiments, the ROIs for brain graphs in the ABIDE and ADHD-200 datasets are defined using the Craddock 200 atlas, while in the ADNI dataset, ROIs are based on the AAL atlas. Consequently, the ABIDE, ADHD-200, and ADNI datasets contain **200, 190, and 90 ROIs**, respectively. We revised Section 4.1 by claiming the number of nodes in each dataset. You may refer to lines 306-307.
>
> The current parcellations we selected (90, 190, and 200 ROIs) are commonly used in brain graph studies. Particularly, the AAL atlas is a widely accepted standard for small parcellation, providing 90 regions. Parcellations with fewer than 90 ROIs, such as the Desikan-Killiany atlas with 68 ROIs, are used less frequently because they may **reduce spatial resolution** and might not capture fine-grained functional information [1,2]. Therefore, smaller parcellations are often based more on structural anatomy than functional connectivity [3].  For this reason, **we consider 90 ROIs as the small parcellation in this study**. Our results on the ADNI dataset demonstrate that BioBGT is effective even with this small parcellation, further validating its performance across varying levels of parcellations.
>
> We appreciate your suggestion to explore the impact of smaller parcellations. In future work, we will try smaller parcellation on brain structural connectivity (such as brain graph based on DTI data).
>
> [1] de Reus, M. A., & Van den Heuvel, M. P. (2013). The parcellation-based connectome: limitations and extensions. Neuroimage, 80, 397-404.
>
> [2] Desikan, R. S., Ségonne, F., Fischl, B., Quinn, B. T., Dickerson, B. C., Blacker, D., ... & Killiany, R. J. (2006). An automated labeling system for subdividing the human cerebral cortex on MRI scans into gyral based regions of interest. Neuroimage, 31(3), 968-980.
>
> [3] Hagmann, P., Cammoun, L., Gigandet, X., Meuli, R., Honey, C. J., Wedeen, V. J., & Sporns, O. (2008). Mapping the structural core of human cerebral cortex. PLoS biology, 6(7), e159.
>
> **Weakness 2**: Even though the ablation study makes a good job in comparing different alternatives to the node importance encoding by comparing other centrality measures, there is no ablation study for the community contrastive strategy.
>
> **Response**: Thank you for your comments. We’d like to clarify that the ablation studies include two parts: (1) removing node important encoding and (2) removing functional module-aware self-attention. Particularly, our functional module-aware self-attention mechanism includes two steps: the community contrastive strategy-based functional module extractor and an updated self-attention. You may refer to Section 3.2. Therefore, to verify the effectiveness of our functional module-aware self-attention, we **remove the community contrastive strategy-based functional module extractor** and replace our FM-Attn(∙) with a normal self-attention Attn(∙), denoted as “-FM-Attn(∙)”. You may refer to Section 4.3.
>
> **Weakness 3**: The work lacks comparison with more "traditional" ML models like random forests or SVM. In a few fMRI works/papers that I have seen in the past, and also based on my experience with these datasets, I typically see is that more "traditional" models achieve similar or even better performance than deep learning when trained on the upper-triangle of the correlation matrices (which has been historically widely explored in the connectomics field). In this sense, for a conference like ICLR, I find the lack of comparisons with traditional ML models in this specific context a weakness of this paper.
>
> **Response**: Thank you for your valuable suggestion. We have added SVM and Random Forest as our baselines. You may refer to Tables 1, 6, 7, and 8 for the experimental results.

---

> ### Author Response · Authors · 2024-11-20
> **Response to Reviewer drzD (Part 2)**
>
> **Weakness 4**: This work has an entire section (4.4) focused on the biological plausibility of its model, but these results do not discuss whether these modules correspond to any known brain structure or medical knowledge. Furthermore, it is not clear how these different modules contribute to the final classification, and thus it is not clear how plausible and good these representations are.
>
> **Response**: Thank you for your insightful comments. We realize that the displayed results (heatmaps of the average self-attention scores) failed to illustrate the biological plausibility of our functional module-aware self-attention, because no functional module labels were provided. This makes our results not convincing enough. We’ve revised Figure 5 by displaying empirical labels of brain regions and functional modules. The empirical labels are provided based on Dosenbach et al [4, 5].  ROIs are classified into 6 functional modules, including visual cortex (Vis), motion control (MC), cognition control (CC), auditory cortex (Aud),  language processing (LP), and executive control (EC). Please see Table 9 in Appendix D.4 for the detailed functional module division. The updated Figure 5 now displays average self-attention scores with these functional module labels applied. As seen in Figure 5, compared with other methods, the learned attention scores of our model align the division of functional modules better. For example, nodes within the visual cortex and motion control exhibit higher attention similarity. This indicates that our model achieves a degree of biological plausibility.
>
> Regarding the labels and heatmaps, we’d like to give more explanation:
> - The functional modules used in this study are based on empirical labels, which are not definitive boundaries. These labels represent the best effort to categorize regions based on known functional associations, but they are inherently limited due to the complex biological properties of the brain graph. Perfect alignment between our model’s output and these empirical labels is challenging due to these limitations.
> - Empirical functional modules often encompass ROIs from diverse brain regions, resulting in heterogeneity within each module. For example, in the auditory cortex, both temporal lobe regions (e.g., 'temporal 103' and 'temporal 95') and thalamic regions (e.g., 'thalamus 57' and 'thalamus 58') are included due to their involvement in auditory processing [6,7]. This diversity may reduce the uniformity of high self-attention scores within the module.
> - Because of limited label availability, there are no available labels for the atlas of ADNI and ABIDE datasets. Therefore, we can only provide labels for the ADHA-200 dataset.
>
> We have revised Section 4.5 (you may refer to lines 442-452) and provided the detailed discussion in Appendix D.4. These two parts are highlighted with purple text.
>
> [4] Dosenbach, N. U., Nardos, B., Cohen, A. L., Fair, D. A., Power, J. D., Church, J. A., ... & Schlaggar, B. L. (2010). Prediction of individual brain maturity using fMRI. Science, 329(5997), 1358-1361.
>
> [5] Dosenbach, N. U., Fair, D. A., Miezin, F. M., Cohen, A. L., Wenger, K. K., Dosenbach, R. A., ... & Petersen, S. E. (2007). Distinct brain networks for adaptive and stable task control in humans. Proceedings of the National Academy of Sciences, 104(26), 11073-11078.
>
> [6] Jones, E. G. (2012). The thalamus. Springer Science & Business Media.
>
> [7] Rauschecker, J. P., & Scott, S. K. (2009). Maps and streams in the auditory cortex: nonhuman primates illuminate human speech processing. Nature Neuroscience, 12(6), 718-724.

---

> ### Author Response · Authors · 2024-11-20
> **Response to Reviewer drzD (Part 3)**
>
> **Question 1**: Equation 7 mentions "learnable embedding vectors", but I do not see those in the equation. Can the authors specify this better?
>
> **Response**: We apologize if our explanation caused any confusion regarding Equation 7. For node $i$, the NE value, denoted as $\mathcal{NE}(i)$, represents its node importance degree. It is assigned a learnable embedding vector associated with its node importance degree, represented as $\mathbf{x}_{\mathcal{NE}(i)}$. The node embedding is then updated as shown in Equation 7:
>
> $$
> \mathbf{x’}_i=\Phi(\mathbf{x}_i) = \mathbf{x}_i + \mathbf{x}_{\mathcal{NE}(i)}
> $$
>
> **Question 2**: Did the authors try other contrastive loss functions beyond the one in equation 8?
>
> **Response**: Thank you for your comment. We chose InfoNCE as the contrastive loss function, as it is a widely used and well-established approach in contrastive learning. InfoNCE has been shown to effectively maximize mutual information between positive pairs while minimizing similarity between negative pairs, which aligns well with our objective to distinguish functional modules within brain graphs.
>
> **Question 3**: How did the authors calculate the F1/sensitivity/specificity scores for the ADNI dataset, where we have 3 classification labels?
>
> **Response**: For the ADNI dataset, we calculate the F1, sensitivity, and specificity scores independently for each class. Then, we calculate the average of these values across all classes to provide the evaluation of model performance in the multiclass setting.
>
> **Question 4**: In Conclusion, the authors argue that they "managed to keep the number of parameters comparable to other models". Where is this shown in the paper?
>
> **Response**: We apologize for not including a direct comparison of parameter numbers in the main text. To address this, we have added a parameter number comparison between our model and some baseline models in the Appendix (please see Table 4 in Appendix D.2). This addition demonstrates that BioBGT maintains a parameter number comparable to other models. Thank you for highlighting this point.
>
> Thanks again for your comments. Hope we have well addressed your concerns. If there are any other issues remaining, we are pleased to address them further.

---

> ### Comment · Reviewer_drzD · 2024-11-20
>
> I thank the reviewers for their time answering and tackling my questions point-by-point. I'll comment on each point below.
>
> **Weakness 1:**
> I appreciate the further clarification, which definitely solves my issue. As the authors only mentioned the "AAL" atlas and referenced the AAL3 paper for the ADNI data, I was expecting more ROIs than just 90. Maybe this means the paper referenced chosen should not have been the AAL3 one?
>
> **Weakness 2:**
> I apologise for my confusion, I understand how my identified weakness is not an actual one.
>
> **Weakness 3:**
> I thank the authors for this inclusion. It's not specifically mentioned in the paper, but I'm guessing they've used the upper-triangle of the correlation matrices, is that correct? I have to admit I find the results with SVM/RF a bit smaller than expected, but the code shared by the authors doesn't seem to have the implementation details for these models, so it's difficult for me to check. This highlights another small weakness of this work in which it is not mentioned (in the paper and in the code) how the hyperparameters were selected for the other models, and it is well known how much hyperparameters can influence final performance. I'm not sure this can be tackled over the remaining rebuttal period, but I'd just ask the authors to at least mention which kernel was used for the SVM in their final paper version (ie, linear, poly, rbf?) for clarity.
>
>
> **Weakness 4:**
> The inclusion of the empirical labels in Figure 5 definitely helps clarify the paper's claims, and makes the points defended by the authors much clearer. For a computational conference like ICLR, I believe this is enough to justify biological plausibility, but hopefully reviewer 75RC will be able to answer the author's rebuttal, as this reviewer seemed to have a stronger opinion on this.
>
>
> **Question 1:**
> To further explain myself, somehow I understood that the "learnable embedding vectors" were about *learnable parameters* of the network, but after reading the authors rebuttal I understand I confused things, and obviously the authors just meant representation learning, apologies.
>
> **Question 2:**
> Thanks for the answer, I understand that InfoNCE is a good choice, I was just asking whether you've tried other contrastive losses. This is obviously not a big issue, it was just for my own clarity to understand the limits of the experiments conducted in this paper.
>
> **Question 3:**
> If understand correctly, this means the authors used what is commonly used as "macro averaging", right? I recommend the authors to make it clear somewhere in the paper, as there are different strategies for reporting multiclass classification metrics.
>
> **Question 4:**
> Thanks, I consider my question to be answered.

---

> > ### Author Response · Authors · 2024-11-21
> > **Response to follow-up comment by Reviewer drzD**
> >
> > **Weakness 1**
> >
> > **Response**: We apologize for not making this clear in the previous version of the manuscript. You are absolutely correct that the reference to the AAL3 paper may be misleading, as AAL3 includes a larger number of brain areas compared to the original AAL atlas, which contains 90 ROIs. To address this, we have replaced the reference to the AAL3 paper with the reference of the original AAL atlas [1]. You may refer to line 303 and lines 705-708.
> >
> > [1] Tzourio-Mazoyer, N. et al. (2002). Automated anatomical labeling of activations in SPM using a macroscopic anatomical parcellation of the MNI MRI single-subject brain. Neuroimage, 15(1), 273-289.
> >
> > **Weakness 3**
> >
> > **Response**: Thank you for your further comment. For SVM, we use the linear kernel. Here, we’d like to provide the implementation details for SVM and RF:
> > | SVM       | RF       |
> > |----------------|----------------|
> > | Regularization parameter: 1   | The number of trees: 100   |
> > | Kernel: linear   | The maximum depth: None   |
> > |   | Min_samples_split: 2   |
> > |   | min_samples_leaf: 1  |
> > |   | max_features: sqrt  |
> >
> > Following your comments, we have revised the paper to include the kernel information of SVM. You may refer to line 328.
> >
> > We have tried our best to compare our SVM and RF results with those reported in papers that trained these models on the same brain data. And we found, [2] trained RF on the ADNI dataset, and [3] trained SVM on the ADNI dataset. However, both papers focus on binary classification (NC vs MCI), and their dataset sizes are a bit different from ours. The results reported in [2] for RF are 57.60% ACC and 55.25% AUC, while [3] reported 52.2% ACC for SVM. To compare with them, we have also trained SVM and RF for binary classification (NC vs MCI) on our ADNI dataset. Our results show that SVM achieved 50.00% ACC, which falls within an acceptable margin of error compared to the results in [3]. Similarly, RF achieved 58.82% ACC and 59.45% AUC, both of which are within an acceptable margin of error compared to the results in [2].
> >
> > In addition, to enhance the transparency of our work, we have submitted the code for SVM and RF, as well as the dataset (ADNI) used in this paper, in the supplementary material. All experimental details will be provided after the paper is published to ensure transparency and reproducibility.
> >
> > [2] Dong, Z., et al. (2023, October). Beyond the snapshot: Brain tokenized graph transformer for longitudinal brain functional connectome embedding. In International Conference on Medical Image Computing and Computer-Assisted Intervention.
> >
> > [3] Yang, Y., et al. (2023). Mapping multi-modal brain connectome for brain disorder diagnosis via cross-modal mutual learning. IEEE Transactions on Medical Imaging.
> >
> > **Question 2**
> >
> > **Response**: Thank you for your kind reply. Actually, I haven’t tried other contrastive losses in this study. However, we fully acknowledge that exploring alternative contrastive losses could provide further insights into the limitations of our approach. We appreciate your suggestion, which offers great inspiration for our future work.
> >
> > **Question 3**
> >
> > **Response**: Yes, we did use macro averaging. To make this clearer in the paper, we have added an explanation (please refer to lines 320-324). Thank you for your helpful suggestion.

---

> > > ### Comment · Reviewer_drzD · 2024-11-21
> > >
> > > I thank the authors for their further clarifications. I have to admit that SVM with a linear kernel might not be the fairest choice for comparison, but I reckon at this stage, and with the results from the RF, likely it wouldn't change much. I also appreciate the comparison of the authors' results regarding SVM and RF in the broader literature.
> > >
> > > I don't have any further comments or suggestions, and thus will wait for the remaining reviewers to answer back the author's rebuttal. Thanks for such prompt and detailed replies.

---

> > > > ### Author Response · Authors · 2024-11-24
> > > >
> > > > We are glad to hear that our rebuttal has addressed your concerns and questions. Your insightful suggestions greatly help us improve the quality of our paper. It is truly valuable for us to have such a meaningful discussion with you.
> > > >
> > > > Thank you very much for your time and effort in reviewing our paper and engaging in further discussions with us.

---

> > > > > ### Comment · Reviewer_drzD · 2024-11-27
> > > > >
> > > > > I appreciate the answers and discussions that the authors had with me and the other reviewers. Based on all this I'm increasing the score from 6 to 8 (accept), as well as increasing the contribution score from 3 to 4. There are still some small weaknesses that I've mentioned in this discussion and that is why I do not increase the score of this paper to the maximum value. I thank the authors for the insightful discussions and the time taken to address my concerns.

---

> > > > > > ### Author Response · Authors · 2024-11-27
> > > > > > **Thank you so much!**
> > > > > >
> > > > > > Thank you very much for raising your rating, as well as the contribution score. We really appreciate your recognition of our work. It would be a great motivation for us to dedicate ourselves to doing research in this field.
> > > > > >
> > > > > > We also greatly appreciate the chance to have such a meaningful discussion with you. Your profound insights and valuable suggestions have significantly helped us improve our work.
> > > > > >
> > > > > > Thank you so much!

---

### Official Review · Reviewer_75RC · 2024-11-03

**Soundness:** 2
**Presentation:** 2
**Contribution:** 2
**Rating:** 5
**Confidence:** 4

**Summary:**

This paper proposes BioBGT, a framework that incorporates node importance encoding and community-aware Graph Transformer for learning the network representation of the brain.
Strengths of this work include that 1) the proposed method is motivated by incorporating prior knowledge of the brain, and 2) it shows clear improvement over the presented baselines.
Some concerns and suggestions are outlined in the `Questions` section.

**Strengths:**

- Motivation to incorporate prior knowledge of the brain into the framework
- The proposed method shows clear improvement over the presented baselines

**Weaknesses:**

- The authors’ emphasis on biological plausibility is weakly supported both theoretically and experimentally.
- The baseline model performance in the experiments is generally lower than expected, so verification may be necessary.

**Questions:**

### Major concerns
- The biological properties of the brain graph are very complex and not much known yet. Strong assumptions and statements about "biologically plausible" properties can be generally toned down throughout the manuscript.
- The proposed methodology does not seem to significantly incorporate the biological property of the human brain as the authors claim, given that the theoretical and empirical findings are not very supportive. Specifically:
  - 1) In Section 4.4, using PCC values (which seem to reflect the average functional connectivity strength across the nodes) as a representative measure of biological plausibility is logically tenuous. Even if we assume that the PCC value is representative of biological plausibility, the quantitative similarity between NE values and PCC values has not been studied.
  - 2) The BioBGT output demonstrated in Figure 5 (c) does not seem to represent a biologically plausible network, which deviates from the modular patterns of brain functional connectivity.
- Section 4.5 focuses more on generalizability across datasets than on comparing the scale of the model parameter size, data, and computation. Please consider revising the possibly misleading term 'scalability' to 'generalizability'.
- The generalizability of BioBGT to non-biological networks shown in Section 4.5 seems to contradict the authors' emphasis on the strength of the BioBGT, that it introduces 'biological' property of the human brain to the framework. Please revise the logical consistency of Section 4.5 since it reads as if any network that contain hubs and modules are considered 'biological'.
- Merging Tables 1, 2, and 6 and reducing the number of presented metrics (e.g., AUC and ACC) would improve the readability of the results.
- Ablation studies seem to show redundant node importance ablations with various graph metrics. Please consider reducing the number of node importance ablations and merging the result with FM-Attn ablation results.
- In Eq. (7), please elaborate on the rationale for incorporating node importance through addition rather than multiplication or concatenation.



### Minor concerns
- Clarification needed on “high correlation” (line 054): please clarify if this refers to the correlation of temporal changes.
- The category "Brain Graph Learning Models" can be revised to "Message-Passing Neural Networks (MPNNs)" or "Graph Neural Networks (GNNs)". GAT is classified as a Graph Transformer, but it is more often considered as a GNN. Please consider re-categorizing the model.
- Please consider adding more recent Graph Transformer-based studies on functional connectivity as baseline methods.


### Recommendation
Given the above concerns, especially the weakness of the experimental results in supporting the authors' claim, I initially recommend reject of the paper.


### Recommendation after rebuttal
Given the authors' response to the concerns, I recommend weak reject of the paper.

---

> ### Author Response · Authors · 2024-11-20
> **Response to Reviewer 75RC (Part 1)**
>
> Thank you very much for your valuable feedback on our paper. We fully understand your concerns and very are pleased to address your questions. Based on your comments, we have revised the manuscript (red text in the manuscript) and provide detailed responses below.
>
> **Weakness 1**: The authors’ emphasis on biological plausibility is weakly supported both theoretically and experimentally.
>
> **Response**: Please see the response to Major concern 2.1 and 2.2.
>
> **Weakness 2**: The baseline model performance in the experiments is generally lower than expected, so verification may be necessary.
>
> **Response**: Thank you for your comments. To conduct experiments, each dataset is randomly split, with 80% used for training, 10% for validation, and 10% for testing. All experimental results are the average values of 10 random runs on test sets with the standard deviation. We have compared our reproduced results with the original papers, which used the same datasets:
> - In the original BRAINNET paper, which also conducted experiments on the same ABIDE dataset, the reported accuracy was 71.0%$\pm$ 1.2%. Our reproduction of this result achieved an accuracy of 68.24%$\pm$ 2.24%, which is within a reasonable margin of error. Notably, BioBGT achieved an accuracy of 74.00%$\pm$ 2.01%, outperforming the results presented in the original BRAINNET paper.
> - In the original GroupBNA paper, which conducted experiments on the ABIDE dataset, the reported accuracy was 60.56%. Our reproduction of this result achieved an accuracy of 63.14%$\pm$ 2.65%, which is a reasonable increase within the margin of experimental variability.
>
> Please note that due to the randomness of experimental results, the reproduced results of some baselines may be a little different from those in the original papers.
>
> **Major concern 1**: The biological properties of the brain graph are very complex and not much known yet. Strong assumptions and statements about "biologically plausible" properties can be generally toned down throughout the manuscript.
>
> **Response**: We do agree that strong claims about "biologically plausible" properties should be toned down, given the ongoing discoveries in brain science. We have thoroughly reviewed the manuscript to soften these statements, ensuring they align closely with current knowledge. In particular, we have revised:
> - Lines 73-74 “We aim to enhance the biological plausibility of the learned representations, with a focus on encoding the small-world architecture of brain graphs. ” to “*We aim to improve the alignment of the learned representations with biological properties, particularly by encoding small-world features commonly observed in brain graphs.*”
> - Lines 75-77 “We propose a Biologically Plausible Brain Graph Transformer (BioBGT), which emphasizes the biological plausibility of brain graph representations from two perspectives: node importance encoding and functional module encoding.”  to “*We propose a Biologically Plausible Brain Graph Transformer (BioBGT), which aligns brain graph representations with biological properties through two main components: node importance encoding and functional module encoding*.”
> - Lines 89-90 “This paper highlights that brain graph representations obtained from learning models should be biologically plausible.” to “*This paper highlights that brain graph representations obtained from learning models should align closely with the biological properties of the brain*.”
> - Lines 107-108 “This paper claims that the biological plausibility of brain graph representations is reflected in the representations of these two indicators.” to “*This paper suggests that the biological plausibility of brain graph representations can be reflected in the representations of these two indicators.*”
> - The conclusion by providing explanations: “*While these findings are encouraging, some limitations remain. Firstly,  according to current neuroscience knowledge, the biological properties of the brain are highly complex and remain uncertain, with many underlying mechanisms still requiring further research. Therefore, it is unlikely that a fully biologically plausible brain graph can be constructed. Instead, we can strive to build brain graphs that are as biologically plausible as possible, drawing on existing knowledge, such as the brain's small-world architecture.*” You may refer to lines 512-517.

---

> ### Author Response · Authors · 2024-11-20
> **Response to Reviewer 75RC (Part 2)**
>
> **Major concern 2.1**: In Section 4.4, using PCC values (which seem to reflect the average functional connectivity strength across the nodes) as a representative measure of biological plausibility is logically tenuous. Even if we assume that the PCC value is representative of biological plausibility, the quantitative similarity between NE values and PCC values has not been studied.
>
> **Response**: Thank you for your comments. PCC has been widely used to approximate brain functional connectivity strength, and it serves as a biologically reliable metric for measuring global brain structure. For example, [1] uses PCC values as node features in brain graphs, reflecting the structural information of nodes. [2] uses PCC values to estimate the connectivity of brain regions. [3] provides evidence that PCC-based brain graph structures exhibit greater reliability than those measured by other metrics. They claim that **PCC is a suitable choice for measuring the global topological properties of functional brain graphs.** A high PCC value between two nodes indicates a strong correlation, suggesting these nodes may interact closely within the brain graph. Thus, a node's average PCC value can indicate its "communication strength" with the rest of the graph, which can be interpreted as an aspect of its **structural importance** in the graph.
>
> The NE values capture structural characteristics relevant to biological properties, specifically measuring node importance in global topology and information propagation. **A higher NE value indicates that the node plays a more essential role in information propagation and has strong correlations with other nodes, aligning with the strong correlations represented by PCC.** Therefore, a node with a larger NE value also tends to have a relatively high PCC value. NE’s assessment of node importance aligns with the biologically plausible connectivity captured by PCC.
>
> Compared to PCC, NE values provide a clearer differentiation between nodes in terms of importance. As shown in Figures 4, 8, 9, and 10, the trends in NE and PCC values are similar, but NE values have greater variance, providing a more distinct measure of node importance and making it easier to identify critical nodes within the graph.
>
> We hope our response well addresses your concern about the similarity between NE values and PCC values. We have revised the manuscript by providing more detailed explanations. You may refer to lines 424-434.
>
> [1] Li, X., Zhou, Y., Dvornek, N., Zhang, M., Gao, S., Zhuang, J., ... & Duncan, J. S. (2021). Braingnn: Interpretable brain graph neural network for fmri analysis. Medical Image Analysis, 74, 102233.
>
> [2] Pedersen, M., Zalesky, A., Omidvarnia, A., & Jackson, G. D. (2018). Multilayer network switching rate predicts brain performance. Proceedings of the National Academy of Sciences, 115(52), 13376-13381.
>
> [3] Liang, X., Wang, J., Yan, C., Shu, N., Xu, K., Gong, G., & He, Y. (2012). Effects of different correlation metrics and preprocessing factors on small-world brain functional networks: a resting-state functional MRI study. PloS One, 7(3), e32766.

---

> ### Author Response · Authors · 2024-11-20
> **Response to Reviewer 75RC (Part 3)**
>
> **Major concern 2.2**: The BioBGT output demonstrated in Figure 5 (c) does not seem to represent a biologically plausible network, which deviates from the modular patterns of brain functional connectivity.
>
> **Response**: Thank you very much for your comments. We recognize that the heatmaps in the original manuscript may not clearly demonstrate the biological plausibility of our functional module-aware self-attention mechanism. Figure 5 did not provide functional module labels, which may make the visualization unclear. To address this, we have revised Figure 5 to display average self-attention scores with available functional module labels applied. Based on the empirical labels provided by Dosenbach et al. [4,5], the ROIs are classified into 6 functional modules: visual cortex (Vis), motion control (MC), cognition control (CC), auditory cortex (Aud), language processing (LP), and executive control (EC). For more details, please refer to Table 9 in the Appendix D.4.
>
> As seen in Figure 5, compared with other methods, the learned attention scores of our model align the division of functional modules better. For example, nodes within the visual cortex exhibit higher attention similarity, which suggests that our model captures some functional modularity in brain graphs. However, the alignment patterns might not appear perfectly consistent. There are several reasons:
> - The brain's functional connectivity is highly complex, with extensive interactions between regions, which makes it difficult to fully capture functional modules. Our current knowledge about their exact boundaries and connectivity patterns is still evolving.
> - The functional modules are based on empirical labels, which are not definitive boundaries. These labels represent the best effort to categorize regions based on known functional associations, but they are inherently limited due to the complex biological properties of the brain graph. Perfect alignment between our unsupervised self-attention output and these empirical labels is challenging due to these limitations.
> - Empirical functional modules often encompass ROIs from diverse brain regions, resulting in heterogeneity within each module. For example, in the auditory cortex, both temporal lobe regions (e.g., 'temporal 103' and 'temporal 95') and thalamic regions (e.g., 'thalamus 57' and 'thalamus 58') are included due to their involvement in auditory processing [6,7]. This diversity may reduce the uniformity of high self-attention scores within the module. The block-like patterns of Aud are less pronounced because auditory modules span multiple brain regions, unlike visual cortex and motion control, which mainly contain ROIs from the occipital and cerebellar regions, respectively.
>
> While our results may not perfectly match all functional modules, they do achieve a degree of biological plausibility. Compared to other methods, our model demonstrates clearer differentiation between some functional modules. Achieving a fully biologically plausible graph is practically unfeasible, but our approach provides a step in that direction. We have revised Section 4.5 (you may refer to lines 442-452) and provided detailed discussion in Appendix D.4. There two parts are highlighted with purple text.
>
>  [4] Dosenbach, N. U., Nardos, B., Cohen, A. L., Fair, D. A., Power, J. D., Church, J. A., ... & Schlaggar, B. L. (2010). Prediction of individual brain maturity using fMRI. Science, 329(5997), 1358-1361.
>
> [5] Dosenbach, N. U., Fair, D. A., Miezin, F. M., Cohen, A. L., Wenger, K. K., Dosenbach, R. A., ... & Petersen, S. E. (2007). Distinct brain networks for adaptive and stable task control in humans. Proceedings of the National Academy of Sciences, 104(26), 11073-11078.
>
> [6] Jones, E. G. (2012). The thalamus. Springer Science & Business Media.
>
> [7] Rauschecker, J. P., & Scott, S. K. (2009). Maps and streams in the auditory cortex: nonhuman primates illuminate human speech processing. Nature neuroscience, 12(6), 718-724.
>
> **Major concern 3**: Section 4.5 focuses more on generalizability across datasets than on comparing the scale of the model parameter size, data, and computation. Please consider revising the possibly misleading term 'scalability' to 'generalizability'.
>
> **Response**: Thank you for your suggestion. We realize that the term “scalability” is misused here. Following your valuable suggestion, we have revised “scalability” to “generalizability”.

---

> ### Author Response · Authors · 2024-11-20
> **Response to Reviewer 75RC (Part 4)**
>
> **Major concern 4**: The generalizability of BioBGT to non-biological networks shown in Section 4.5 seems to contradict the authors' emphasis on the strength of the BioBGT, that it introduces 'biological' property of the human brain to the framework. Please revise the logical consistency of Section 4.5 since it reads as if any network that contain hubs and modules are considered 'biological'.
>
> **Response**: The goal of Section 4.5 was to illustrate BioBGT's generalizability beyond brain graphs to other networks with similar structural properties, specifically the presence of hubs and modules. However, inspired by your comments, we realize that this section may depart from the primary focus of BioBGT. It shouldn’t be in the main part of this paper. We have revised this section and moved it to Appendix D.5 as supplementary evidence of BioBGT's generalizability. We apologize for any confusion this may have caused and appreciate your suggestions.
>
> **Major concern 5**: Merging Tables 1, 2, and 6 and reducing the number of presented metrics (e.g., AUC and ACC) would improve the readability of the results.
>
> **Response**: Thank you for this suggestion. We have merged Tables 1, 2, and 6 with presenting metrics ACC and AUC. Please refer to Table 1 in the updated manuscript. Tables 5, 6, and 7 in Appendix D.3 provide the results of the other three metrics on three datasets, respectively.
>
> **Major concern 6**: Ablation studies seem to show redundant node importance ablations with various graph metrics. Please consider reducing the number of node importance ablations and merging the result with FM-Attn ablation results.
>
> **Response**:  Following your suggestion, we have merged node importance encoding ablation and functional module-aware self-attention ablation into Section 4.3. The results of BioBGT and its altered models removing (1) node importance encoding (-NE) and (2) FM-*Attn* (-FM-*Attn*) are compared in the updated Figure 3.
>
> The comparison between NE and other node importance measurement methods is moved to a new section (Section 4.4: Comparative Analysis of Node Importance Measurement).
>
> **Major concern 7**: In Eq. (7), please elaborate on the rationale for incorporating node importance through addition rather than multiplication or concatenation.
>
> **Response**: Thank you for this comment. We believe that addition is an efficient and stable way to incorporate node importance encoding into node representations. Furthermore, adding node importance directly to the node representations stands as an enhancement within the feature space without fundamentally altering the structure of their representations.
> - Using multiplication to encode node importance could amplify or diminish some feature dimensions, which might distort the representation. Especially if node importance values vary widely, multiplication could lead to instability due to excessive feature scaling.
> - Concatenating node importance with the original features effectively doubles the feature dimensionality, increasing the computational complexity and potentially leading to overfitting.

---

> ### Author Response · Authors · 2024-11-20
> **Response to Reviewer 75RC (Part 5)**
>
> **Minor concern 1**: Clarification needed on “high correlation” (line 054): please clarify if this refers to the correlation of temporal changes.
>
> **Response**: Thank you for your comment. In functional connectivity, the connection between two ROIs represents the correlation between their respective time series. Therefore, the strong connection between two ROIs reflects a high temporal correlation, indicating that these regions are functionally associated.
>
> **Minor concern 2**: The category "Brain Graph Learning Models" can be revised to "Message-Passing Neural Networks (MPNNs)" or "Graph Neural Networks (GNNs)". GAT is classified as a Graph Transformer, but it is more often considered as a GNN. Please consider re-categorizing the model.
>
> **Response**: Thank you for this valuable suggestion. In the revised manuscript, we re-categorize the baselines into the following categories: (1) Machine learning methods; (2) Graph transformer models; (3) Graph neural networks.
>
> **Minor concern 3**: Please consider adding more recent Graph Transformer-based studies on functional connectivity as baseline methods.
>
> **Response**: We have added three recent graph transformer-based studies as the baselines, including [8], [9], and [10]. Please refer to Tables 1, 6, 7, and 8 for the experimental results.
>
> [8] Liu, C., Yao, Z., Zhan, Y., Ma, X., Pan, S., & Hu, W. (2024). Gradformer: Graph Transformer with Exponential Decay. In Proceedings of the Thirty-Third International Joint Conference on Artificial Intelligence, Jeju, South Korea, August 3-9, pp. 2171–2179, 2024
>
> [9] Liu, C., Zhan, Y., Ma, X., Ding, L., Tao, D., Wu, J., ... & Du, B. (2024). Exploring sparsity in graph transformers. Neural Networks, 174, 106265.
>
> [10] Deng, C., Yue, Z., & Zhang, Z. (2024). Polynormer: Polynomial-expressive graph transformer in linear time. In Proceedings of the International Conference on Learning Representations, 2024
>
> Thanks again for your comments. Hope we have well addressed your concerns. If there are any other issues remaining, we are pleased to address them further.

---

> > ### Comment · Reviewer_75RC · 2024-11-22
> > **Response to the authors**
> >
> > I sincerely appreciate the authors' prompt and detailed response to my comments. They addressed my concerns significantly.
> > However, in general, I am afraid that I am still not convinced about the paper's
> > 1) underlying assumption on the statement 'small-worldness is biological plausibility', and
> > 2) the presentation and claims that deviate from the knowledge of neuroimaging and fMRI signals.
> >
> > Even if an agreement is made on statement 1, the experiments still do not confirm that the proposed method promotes small-worldness $\sigma$ of the brain graph.
> >
> > ---
> >
> > **Major Concern 2.1**
> > - Please consider revising the term 'PCC' to 'Functional Connectivity (FC)' for clarity, which is a more widely used term in the neuroimaging field.
> > - The claim that the average FC across nodes represent 'communication strength' still does not seem biologically plausible, given the characteristics of fMRI signal acquisition. For example, global signal regression can significantly affect the average FC strength, which is a preprocessing step in controversy. Average FC is a value that can be confounded by various factors, and the authors should consider other representative graph metrics for 'communication strength', instead of the average FC.
> >
> > **Major Concern 2.2**
> > - Modular patterns seen in Figure 5c still seems trivial, given that the modulaty is an inherent pattern of FC from fMRI data. The interpretation can further include how the attentions of each intrinsic connectivity networks (or modules) are actually related to the biological patterns of the target illness, here the ADHD. Also, please consider interpreting the attention patterns between the three datasets, and confirm the patterns follow the neuroscientific evidences of ASD, ADHD, and Alzheimer's disease to further support the claims.
> >
> > **Major Concern 4.**
> > - I appreciate the authors' response. The underlying assumption that 'small-worldness is biological plausibility' of this paper seems to be an overshoot and harming the strength of this paper consistently, as I commented above.
> >
> > **Major Concern 7.**
> > - The claim on 'stability' of the addition could be further elaborated, since adding the two features might diminish the information from the two features. Please consider supporting the claim either theoretically or empirically.
> >
> > **Minor Concern 1.**
> > - Please consider revising the term 'PCC' to 'functional connectivity (FC)' as suggested above.

---

> > > ### Author Response · Authors · 2024-11-23
> > > **Response to follow-up comment by Reviewer 75RC (Part 1)**
> > >
> > > Thank you very much for your response to our rebuttal. We are pleased to provide further clarification and address your concerns. Based on your suggestions, we have revised the manuscript. You may refer to the red and orange text in the updated manuscript.
> > >
> > > ---
> > >
> > >
> > > **Response to Major Concern 2.1**
> > >
> > > We appreciate your suggestions regarding the limitations of average FC strength and acknowledge that it may be influenced by preprocessing steps, such as global signal regression. To address this, we utilize another graph metric, **Node Efficiency (NEff)**, to more appropriately quantify the communication strength of nodes in the brain graph. NEff measures the efficiency of information propagation between a given node and all other nodes in the graph [1]. A high NEff value indicates that the node plays a critical role in information propagation, enabling close and efficient communication with the rest of the network. Also, NEff has been widely utilized in the field of neuroscience to identify critical regions in functional brain graphs [2]. For example, as discussed in [2], the efficiency of a brain graph (including global and local efficiency) describes its capacity for information transfer. Specifically, for an individual node, its NEff reflects its ability to propagate information within the brain graph.
> > > - In the updated manuscript, we have updated **Section 4.5** replacing PCC with node efficiency (NEff). We compare the NE values of each node with their corresponding NEff values to demonstrate the reliability of NE in measuring node importance. You may refer to lines 424-441 and figure 4. Also, in Appendix D.3, we provide the visualizations of NE and NEff values for all nodes in brain graphs across three datasets.
> > > - We have revised and moved the comparison between NE and FC strength to Appendix D.4 as additional material, and it is not included in the main body of the paper.
> > >
> > > Regarding the revision, we would highlight that:
> > > - Node efficiency is based on the shortest path [1], and its calculation depends on the topological structure of the network rather than directly relying on fMRI signals. While node efficiency may still be indirectly influenced by fMRI data preprocessing (e.g., the method used to construct the graph), it focuses more on the graph's topological properties. Therefore, compared to average FC, we believe that node efficiency is more reliable when assessing the structural importance of brain graphs, particularly a node's role in information propagation.
> > > - Due to the density of the curves when all nodes are displayed can make the patterns less discernible, we randomly select 50 nodes for display in the main body to enhance visualization clarity.
> > > - NE values provide a clearer differentiation between nodes in terms of importance. As shown in Figures 4, 8, 9, and 10, the trends in NE and NEff values are similar, demonstrating the reliability of NE. Notably, NE values exhibit greater variance compared to NEff, offering a more distinct measure of node importance. This increased variance enables clearer identification of critical nodes within the graph, further supporting the validity of NE as an effective measure for assessing node importance.
> > >
> > > [1] Latora, V., & Marchiori, M. (2001). Efficient behavior of small-world networks. Physical Review Letters, 87(19), 198701.
> > >
> > > [2] Bullmore, E., & Sporns, O. (2009). Complex brain networks: graph theoretical analysis of structural and functional systems. Nature Reviews Neuroscience, 10(3), 186-198.
> > >
> > > ---
> > >
> > > **Response to Major Concern 2.2**
> > >
> > > Thank you for your valuable suggestions. In Figure 5, we demonstrate that the modular patterns learned by our model better capture the functional module of brain graphs compared to other methods, even in an unsupervised setting. Our results highlight that the learned representations align more closely with the expected modular structures of the brain.
> > >
> > > We understand your concern about the interpretation of functional modules output by our model and appreciate your suggestion that it would be better to analyze the relations between the attention patterns and the target disease. To address this, we have provided the following comparisons:
> > > - Heatmaps of the average self-attention scores of ADHD patients and Normal Control in the ADHD-200 test set (**Figure 14**).
> > > - Heatmaps of the average self-attention scores of ASD patients and Normal Control in the ABIDE test set (**Figure 15**).
> > > - Heatmaps of the average self-attention scores of Normal Control, MCI, and AD patients in the ADNI test set (**Figure 16**).
> > >
> > > The detailed discussion is given in **Appendix D.6** (Analysis of Attention Patterns output by BioBGT).

---

> ### Author Response · Authors · 2024-11-23
> **Response to follow-up comment by Reviewer 75RC (Part 2)**
>
> **Response to Major Concern 4**
>
> Thank you for your insightful comment. We’d like to clarify that the claim of this paper is: “**Brain graph representation learning should preserve the small-world characteristics of the brain, enhancing the biological plausibility of representations to an extent.**” We do not assume or assert that “small-worldness is biological plausibility”.
>
> We do agree with you that it is not correct to equate small-worldness with complete biological plausibility, nor to claim that satisfying small-world characteristics alone guarantees biologically plausible brain graph representation learning. The brain is a highly complex system that continues to be studied and explored.
>
> Based on existing neuroscientific knowledge, **the small-world architecture is one of the typical biological features of the brain** [2-4].
>
> - [2] claims that “*Small-world architectures have been found in several empirical studies of structural and functional brain networks in humans and other animals, and over a wide range of scales in space and time.*”
> - [3] claims that “*One of the earliest and most influential discoveries in network neuroscience was that connectomes are small-world networks.*”
> - [4] claims that “*Human brain structural and functional networks follow small-world configuration.*”
>
> Therefore, we believe that brain graph representation learning should, at a minimum, consider and preserve the brain’s small-world structure to ensure the representations capture an essential aspect of the brain’s biological properties. This can, to some extent, enhance the biological plausibility of brain graph representation learning.
>
> As we describe in the conclusion, it is unlikely that a fully biologically plausible brain graph can be constructed. Our model aims to advance the study of biologically plausible brain graph representation learning. Exploring models that effectively capture the small-world characteristics of the brain is a critical step toward achieving this goal.
>
> We apologize again for any confusion caused by our claims. We have thoroughly reviewed the manuscript and revised the sentences that may not align with our intended statements. You may refer to the text with orange color in the updated manuscript.
>
> [2] Bullmore, E., & Sporns, O. (2009). Complex brain networks: graph theoretical analysis of structural and functional systems. Nature Reviews Neuroscience, 10(3), 186-198.
>
> [3] Seguin, C., Sporns, O., & Zalesky, A. (2023). Brain network communication: concepts, models and applications. Nature reviews neuroscience, 24(9), 557-574.
>
> [4] Liao, X., Vasilakos, A. V., & He, Y. (2017). Small-world human brain networks: perspectives and challenges. Neuroscience & Biobehavioral Reviews, 77, 286-300.
>
>
> ---
>
> **Response to Major Concern 7**
>
> We apologize for not fully addressing your concern in our previous response. (1) Compared to multiplication, addition is a fusion method that **does not involve nonlinear mapping or additional weight parameters**. This avoids the instability that nonlinear methods (e.g., multiplication) might introduce. (2) Compared to concatenation, addition does not alter the dimensionality of the features, and concatenation may double the feature dimensions, potentially increasing computational overhead. Therefore, we summarize our claim that addition is a stable and efficient way to incorporate node importance encoding into node representations.
>
> Many studies have demonstrated the effectiveness of addition as a fusion mechanism. For example:
> - ResNet [5] explicitly highlights that simple addition helps stabilize the training of deep networks and mitigates gradient vanishing issues through linear fusion.
> - In Transformer [6], the output of self-attention is typically added directly to the input. This design ensures stable information propagation while simplifying the fusion process.
> - Graphormer [7] uses addition in its graph representation learning process to integrate structural encoding with node features.
>
> [5] He, K., Zhang, X., Ren, S., & Sun, J. (2016). Deep residual learning for image recognition. In CVPR.
>
> [6] Vaswani, A. (2017). Attention is all you need. NeurIPS .
>
> [7] Ying, C., Cai, T., Luo, S., Zheng, S., Ke, G., He, D., ... & Liu, T. Y. (2021). Do transformers really perform badly for graph representation?. NeurIPS
>
> ---
>
> **Response to Minor Concern 1**
>
> Thank you for your suggestion. We have revised ‘PCC’ to ‘FC' (Appendix D.4).
>
> ---
>
> We really appreciate your expertise and profound insights, particularly in the field of neuroscience. Your valuable suggestions have significantly enhanced the quality of our manuscript. We hope our responses have well addressed your concerns. We are happy to engage in a further discussion with you if any questions or issues remain.

---

> > ### Author Response · Authors · 2024-11-25
> >
> > Dear reviewer 75RC,
> >
> > We really appreciate your suggestions and comments, which are highly valuable to us. As there is only one day remaining for the discussion phase, we want to kindly follow up to ensure there is sufficient time for any further clarifications or concerns you might have. Please let us know if there is anything further we can do to assist or elaborate on.
> >
> > Thank you once again for your time and effort in reviewing our paper and engaging in further discussions with us.
> >
> > Best regards,
> >
> > The authors of submission 6951

---

> ### Comment · Reviewer_75RC · 2024-11-26
> **Response to the authors**
>
> Thank you for the comments.
>
> ---
>
> The authors' follow-up responses were prompt and detailed, which addressed my concerns significantly. I raise my score to weak reject. Still, there exist weaknesses in experimental results (especially low AUC performance / shallow and vague neuroscientific interpretation of follow-up experiments), which serve as the reason for my final decision on the score.

---

> > ### Author Response · Authors · 2024-11-26
> > **Thank you!**
> >
> > Thank you very much for your kind reply and raising your score. Your thoughtful suggestions have greatly helped us improve the quality of our paper.

---

### Official Review · Reviewer_vQGg · 2024-11-04

**Soundness:** 3
**Presentation:** 3
**Contribution:** 2
**Rating:** 6
**Confidence:** 3

**Summary:**

The paper introduces a brain graph representation learning framework aimed at enhancing biological plausibility, with a focus on capturing the small-world architecture of brain networks. Specifically, the authors propose a novel framework called BioBGT, which emphasizes node importance encoding and functional module encoding. The framework consists of two key components; 1) Network entanglement-based node importance encoding, which captures the significance of nodes in the process of information propagation, and 2) Functional module-aware self-attention, which generates node representations that reflect both the functional segregation and integration properties of brain networks. The experimental results demonstrate the effectiveness of this design and its superior performance in brain disease detection tasks.

**Strengths:**

- As emphasized by the authors, their method highlights the biological properties of the brain structure via biological plausibility.
- Overall, the presentations are clear and easy to understand their framework and results.
- The experiments are well-structured with multiple brain datasets, and the proposed model demonstrates strong performance in most cases.

**Weaknesses:**

- The authors propose three main methods: network entanglement-based node importance encoding to reflect the different importance of each node like the hub node, community contrastive strategy-based functional module extractor and an updated self-attention mechanism for functional module-aware self-attention mechanism. However, I couldn’t find much difference from the original works of each methods the authors mentioned they were inspired by. On top of that, the equation 6 and its proof in the appendix needs to be double checked (shouldn’t log2(Z_i Z) be log2(Z_i/Z)?).

- As shown in [1] and the proof in the appendix, the two equations appear identical. However, due to an error in the calculations within the proof, the resulting expressions differ. Therefore, in Eq. (6) of Theorem 1, the second term should be log_2(Z_i/Z) instead of log)_2(Z_iZ). Additionally, while defining the hub characteristics in the brain through node entanglement (NE) seems reasonable, it is difficult to identify significant novelty compared to previous studies.

- Even though the authors provided citations on the quantum entanglement methods, it would be very helpful for the readers to have more details on the definition and why they need to be utilized to quantify the structural information.

- The term “with their biological plausibility preserved” seems to be misleading in Line 345, since the classification performance alone does not directly demonstrate the effect of the biological plausibility, which is never dealt until much later in section 4.4.

- Tables 1 and 2 suggest that the baselines were not compared fairly. Their high standard deviations undermine the credibility of the experimental results.

**Questions:**

- Contributions and novelty need to be clarified as well as the questions raised in the Weakness.

- As seen in [2] and Eq. (9), the self-attention formulations are nearly identical. However, the proposed method claims to enhance functional module encoding by making the exponential kernel trainable. The explanation regarding the benefits of learning the kernel seems insufficient. Could you elaborate on the advantages of this approach? Additionally, the experiments only compare the proposed method with the self-attention mechanism from Eq. (2). A more thorough discussion of the differences between the proposed approach and the method presented in [2] would strengthen the analysis.

- Why did the authors use 50 randomly selected nodes to validate the biological plausibility in node importance encoding? The number of nodes of the brain networks doesn’t seem to be too large. Also, are the results consistent across all datasets?

- In Figure 5, the authors present the average self-attention scores as a heatmap. However, in the results of the proposed BioBGT, the block-like color patterns corresponding to functional modules do not consistently appear across all ROIs. Could you explain the reason for this discrepancy? (For example, between ROIs 40 and 60, the block-like regions with high values are not observed.) Also, I think the caption should be more informative for the figure.

- Moreover,  the biological plausibility in functional module-aware self-attention seems to be ``plausible'' as in its block-like patterns, but do they comply with the neuroscientific prior knowledge? Adding analysis on which functional module match each block may help the biological plausibility of your method.

---

> ### Author Response · Authors · 2024-11-20
> **Response to Reviewer vQGg (Part 1)**
>
> We appreciate your valuable feedback, which is insightful and can improve the quality of our work to a great extent. You definitely pointed out some issues in our paper, giving us inspiration on how to revise the manuscript. In response to your comments and questions, we have provided detailed answers and revised the manuscript accordingly (blue text in the manuscript).
>
> **Weakness 1**: The authors propose three main methods: network entanglement-based node importance encoding to reflect the different importance of each node like the hub node, community contrastive strategy-based functional module extractor and an updated self-attention mechanism for functional module-aware self-attention mechanism. However, I couldn’t find much difference from the original works of each methods the authors mentioned they were inspired by. On top of that, the equation 6 and its proof in the appendix needs to be double checked (shouldn’t $\log_2(Z_i Z)$ be $\log_2(\frac{Z_i}{Z})$)?.
>
> **Response**: In this paper, our BioBGT includes two main components: (1) network entanglement-based node importance encoding and (2) functional module-aware self-attention. The network entanglement-based node importance encoding method aims to capture the varying node importance in information propagation and **encode** this biological property into node representations. The functional module-aware self-attention includes two steps: community contrastive strategy-based functional module extractor and updated self-attention mechanism. This method aims to achieve functional module encoding. We fully understand your concern about the novelty of each method. Here, we’d like to give a detailed explanation:
> -  Our network entanglement-based node importance encoding method addresses the limitations of current methods that overlook the characteristics of brain graphs, particularly different node importance in information propagation. The main contribution of this method is **node importance encoding**. To encode node importance into node representations, we should first measure the importance of each node. Therefore, we apply node entanglement to measure node importance. The original work of node entanglement utilizes network entanglement to measure (or in your word, reflect) node importance. **But in our work, our main proposal is not to reflect/measure node importance, we aim to encode node importance into the representations**. Node importance measurement is just a necessary step for our node importance encoding. In addition, we also conduct experiments on other existing node importance measurement methods (e.g., degree centrality) and compare them with node entanglement (you may refer to Section 4.4). The results demonstrate that node entanglement-based node importance measurement has the best performance. Furthermore, we compare the reliability of node entanglement with other methods in Appendix B. It shows node entanglement can better reflect the importance of a node on information propagation. Therefore, we choose node entanglement to measure node importance.
> - Our functional module-aware self-attention captures the functional segregation and integration characteristics of the brain. The main contribution of this method is **encoding functional modules in an unsupervised manner**.
> - According to your comments, we have double-checked Equation 6 and its proof. We indeed made a mistake in the equation: the term $\log_2(Z_i Z)$ should be $\log_2(\frac{Z_i}{Z})$. We have revised Equation 6 and its proof. We apologize for this mistake and greatly appreciate your feedback.
>
> **Weakness 2**: As shown in Weakness 1 and the proof in the appendix, the two equations appear identical. However, due to an error in the calculations within the proof, the resulting expressions differ. Therefore, in Eq. (6) of Theorem 1, the second term should be log_2(Z_i/Z) instead of log_2(Z_iZ). Additionally, while defining the hub characteristics in the brain through node entanglement (NE) seems reasonable, it is difficult to identify significant novelty compared to previous studies.
>
> **Response**:
> - Thank you for pointing out this mistake.  We have revised Equation 6 and Theorem 1.
> - As we explained in Weakness 1, our novelty is node importance encoding rather than node importance measurement. Measuring node importance through node entanglement is the optimal method before encoding node importance into the representations.

---

> ### Author Response · Authors · 2024-11-20
> **Response to Reviewer vQGg (Part 2)**
>
> **Weakness 3**: Even though the authors provided citations on the quantum entanglement methods, it would be very helpful for the readers to have more details on the definition and why they need to be utilized to quantify the structural information.
>
> **Response**: Thank you for your valuable feedback. Quantum entanglement is a phenomenon in quantum mechanics, describing the correlations between particles [1]. Entangled particles remain connected and exist in a shared quantum state. Mathematically, quantum entanglement is often represented by a **density matrix** of quantum entangled states. **The density matrix captures the interdependences (especially entangled relationships) between particles in the entire entangled system**. When combined with network information theory, concepts from quantum entanglement can provide a powerful lens for analyzing the global topology and information diffusion of graphs [2]. Based on this, we treat the brain graph as an entangled system, where nodes and their connections reflect interdependent states. **The density matrix is used to quantify structural information**. This is because this approach enables us to capture the intricate correlations between nodes, offering insight into both the **global topology** and **information diffusion process** within brain graphs. The density matrix not only encodes the global connectivity patterns but also captures the interdependencies and information flow between nodes.
>
> We do realize current description about quantum entanglement may be not clear enough in the first paragraph of Section 3.1. We have revised this paragraph. You may refer to lines 152-162.
>
> [1] Weedbrook, C., Pirandola, S., García-Patrón, R., Cerf, N. J., Ralph, T. C., Shapiro, J. H., & Lloyd, S. (2012). Gaussian quantum information. Reviews of Modern Physics, 84(2), 621-669.
>
> [2] Huang, Y., Wang, H., Ren, X. L., & Lü, L. (2024). Identifying key players in complex networks via network entanglement. Communications Physics, 7(1), 19.
>
> **Weakness 4**: The term “with their biological plausibility preserved” seems to be misleading in Line 345, since the classification performance alone does not directly demonstrate the effect of the biological plausibility, which is never dealt until much later in section 4.4.
>
> **Response**: Thank you for your suggestion. We agree with you that the term “with their biological plausibility preserved” and also the following sentence “This indicates the small-world architecture of brain graphs is a crucial feature of their representations, reflecting on various node importance degrees (the presence of hubs) and functional modules.” shouldn’t be claimed in Section “Results”. We have revised the sentence as “*The experimental results demonstrate that our model excels in various brain disorder detection tasks.*”. You may refer to line 345 in the updated manuscript.
>
> **Weakness 5**: Tables 1 and 2 suggest that the baselines were not compared fairly. Their high standard deviations undermine the credibility of the experimental results.
>
> **Response**: We fully understand your concern regarding the high standard deviations in our experimental results. Here, we would like to explain that high standard deviations are unavoidable results in brain graph-related experiments, particularly when using ABIDE and ADHD datasets. **This is primarily due to the diverse clinical backgrounds of samples and variations in sampling environments**. For example, the ADHD-200 dataset includes samples from 8 independent imaging sites, and ABIDE aggregates data collected from 24 international brain imaging laboratories. This variability can introduce substantial differences in brain connectivity patterns, which leads to the variability in model performance across runs. To further illustrate this truth, please allow us to give some existing examples from other studies where similar challenges have been encountered:
> - In Table 3 of [3], we can see that their experimental results also demonstrate high standard deviations on ADHD dataset. For example, there are values like 0.600 $\pm$0.215 and 0.612 $\pm$ 0.180.
> - [4] claims that “All reported performances are the average of 5 random runs on the test set with the standard deviation.” In Table 1 of this paper, we can see that the high standard deviation on ABIDE dataset is an unavoidable experimental result. For example, there are values like 51.1%$\pm$40.9%, 53.8%$\pm$41.2%, and 33.9%$\pm$34.2%.
>
> We sincerely ask for your understanding of this unavoidable situation.
>
> [3] Zhao, X., Wu, J., Peng, H., Beheshti, A., Monaghan, J. J., McAlpine, D., ... & He, L. (2022). Deep reinforcement learning guided graph neural networks for brain network analysis. Neural Networks, 154, 56-67.
>
> [4] Kan, X., Dai, W., Cui, H., Zhang, Z., Guo, Y., & Yang, C. (2022). Brain network transformer. Advances in Neural Information Processing Systems, 35, 25586-25599.

---

> ### Author Response · Authors · 2024-11-20
> **Response to Reviewer vQGg (Part 3)**
>
> **Question 1**: Contributions and novelty need to be clarified as well as the questions raised in the Weakness.
>
> **Response**: Please see the response to Weakness 1.
>
> **Question 2**: As seen in [2] and Eq. (9), the self-attention formulations are nearly identical. However, the proposed method claims to enhance functional module encoding by making the exponential kernel trainable. The explanation regarding the benefits of learning the kernel seems insufficient. Could you elaborate on the advantages of this approach? Additionally, the experiments only compare the proposed method with the self-attention mechanism from Eq. (2). A more thorough discussion of the differences between the proposed approach and the method presented in [2] would strengthen the analysis.
>
> **Response**: Thank you for the comments. Equation 2 is the self-attention formulation. Equation 9 is our updated self-attention formulation. I’d like to clarify the difference between them:
> - To calculate our functional module-aware self-attention ($\text{FM-}Attn(\cdot)$), the first step is utilizing community contrastive strategy-based functional module extractor to extract functional modules and obtain **updated functional module-aware representations**. Therefore, compared to standard self-attention, $Attn(\cdot)$ (Equation 2), the input of our $\text{FM-}Attn(\cdot)$ is functional module-aware node representations.
> - We design the self-attention mechanism as a kernel smoother. The trainable exponential kernel is defined on functional module-aware node representations. The kernel function enables self-attention to calculate node similarity at the functional module level, **without disrupting the coherence of functional module-aware node representations** (see Theorem 2). Therefore, it can emphasize relationships within the same functional module while filtering out nodes from different modules. In addition, the kernel provides **flexible and expressive relative positional encoding**, helping the model capture complex relationships between nodes.
>
> Here, we want to highlight that the main contributions of this paper can be summarized in Equation 4. (1) We encode node importance into node representations ($\Phi(\cdot)$). (2) Our functional module-aware self-attention $\text{FM-}Attn(\cdot)$ calculate node similarity at the functional module level.
>
> **Question 3**: Why did the authors use 50 randomly selected nodes to validate the biological plausibility in node importance encoding? The number of nodes of the brain networks doesn’t seem to be too large. Also, are the results consistent across all datasets?
>
> **Response**: Thank you for the comments. We also have tried to use 200 nodes on the ABIDE dataset. When the number of nodes is 200, the trend of the curves becomes dense. Therefore, to enhance visualization clarity, we randomly selected 50 nodes, allowing the trend of the curves to be observed more clearly. In the revised manuscript, we have provided figures with different node numbers across all datasets. The results on all datasets are consistent and demonstrate the biological plausibility in node importance encoding. You may refer to Appendix D.3.
> - For ABIDE dataset, in which each graph has 200 nodes, we visualize the curves for 50 randomly selected nodes (Figure 4) and all 200 nodes (Figure 8).
> - For ADHD-200 dataset, in which each graph has 190 nodes, we visualize the curves for all 190 nodes (Figure 9).
> - For ADNI dataset, in which each graph has 90 nodes, we visualize the curves for all 90 nodes (Figure 10).

---

> ### Author Response · Authors · 2024-11-20
> **Response to Reviewer vQGg (Part 4)**
>
> **Question 4 and Question 5**: In Figure 5, the authors present the average self-attention scores as a heatmap. However, in the results of the proposed BioBGT, the block-like color patterns corresponding to functional modules do not consistently appear across all ROIs. Could you explain the reason for this discrepancy? (For example, between ROIs 40 and 60, the block-like regions with high values are not observed.) Also, I think the caption should be more informative for the figure.
>
> Moreover, the biological plausibility in functional module-aware self-attention seems to be ``plausible'' as in its block-like patterns, but do they comply with the neuroscientific prior knowledge? Adding analysis on which functional module match each block may help the biological plausibility of your method.
>
> **Response**: Thank you for the insightful comments. We recognize that the current heatmaps may not clearly illustrate the biological plausibility of our functional module-aware self-attention. We’ve revised Figure 5 and its caption to provide additional clarity. Taking ADHD-200  dataset as an example, based on the empirical labels of brain regions and functional modules provided by Dosenbach et al [5,6], the ROIs are classified into 6 functional modules, including visual cortex (Vis), motion control (MC), cognition control (CC), auditory cortex (Aud),  language processing (LP), and executive control (EC). Please refer to Table 9 in Appendix D.4 for detailed information. The updated Figure 5 now displays average self-attention scores with these functional module labels applied. As seen in Figure 5, compared with other methods, the learned attention scores of our model align better with the divisions of functional modules. For example, nodes within the visual cortex exhibit higher attention similarity. Some key considerations regarding the results should be noted:
> - The labels are highly dependent on empirical evidence, it remains a challenge to correctly label all ROIs in the brain. Therefore, for the ADHD-200 dataset, we’ve provided the most relevant available labels to highlight functional modules.
> - Empirical functional modules often encompass ROIs from diverse brain regions, resulting in heterogeneity within each module. For example, in the auditory cortex, both temporal lobe regions (e.g., 'temporal 103' and 'temporal 95') and thalamic regions (e.g., 'thalamus 57' and 'thalamus 58') are included due to their involvement in auditory processing [7,8]. This diversity may reduce the uniformity of high self-attention scores within the module. The block-like patterns between ROIs 40 and 60 are less pronounced because auditory and language processing modules span multiple brain regions, unlike visual and motor control, which mainly contain ROIs from the occipital and cerebellar regions, respectively.
> - Because of limited label availability, there are no available labels for the atlas of the ADNI and ABIDE datasets. Therefore, we can only provide labels for the ADHD-200 dataset.
>
> We have revised Section 4.5 (you may refer to lines 442-452) and provided detailed discussion in Appendix D.4. There two parts are highlighted with purple text.
>
> [5] Dosenbach, N. U., Nardos, B., Cohen, A. L., Fair, D. A., Power, J. D., Church, J. A., ... & Schlaggar, B. L. (2010). Prediction of individual brain maturity using fMRI. Science, 329(5997), 1358-1361.
>
> [6] Dosenbach, N. U., Fair, D. A., Miezin, F. M., Cohen, A. L., Wenger, K. K., Dosenbach, R. A., ... & Petersen, S. E. (2007). Distinct brain networks for adaptive and stable task control in humans. Proceedings of the National Academy of Sciences, 104(26), 11073-11078.
>
> [7] Jones, E. G. (2012). The thalamus. Springer Science & Business Media.
>
> [8] Rauschecker, J. P., & Scott, S. K. (2009). Maps and streams in the auditory cortex: nonhuman primates illuminate human speech processing. Nature Neuroscience, 12(6), 718-724.
>
> Thanks again for your comments. Hope we have well addressed your concerns. If there are any other issues remaining, we are pleased to address them further.

---

> ### Author Response · Authors · 2024-11-23
>
> Dear Reviewer vQGg,
>
> As we are now midway through the rebuttal phase, we have further revised the manuscript based on follow-up comments from reviewers drzD and 75RC. As a result, some previously revised sections may have been updated in the latest version of the manuscript. We’d like to highlight the following key updates for your attention:
>
> ---
> **Weakness 3**: The revised content is now located in **lines 149–159**.
>
> ---
>
> **Question 3**: Based on reviewer 75RC’s comments, we have moved the comparison between NE and PCC value to Appdendix D.4. The figures (Figure **11**, **12**, and **13**) with different node numbers across all three datasets can be found in **Appdendix D.4**.
>
> ---
> **Question 4 and Question 5**: A detailed discussion of functional module division is now provided in **Appendix D.5**, with the updated content highlighted in purple text.
>
> ---
> We want to kindly follow up to ensure that our responses have adequately addressed your concerns. Your feedback is highly valued, and we still looking forward to further discussion to clarify or expand on any points as needed. Please feel free to share any additional thoughts or questions you might have.
>
> Thank you once again for your time and effort in reviewing our paper.

---

> > ### Author Response · Authors · 2024-11-26
> >
> > Dear reviewer vQGg,
> >
> > We really appreciate your effort in reviewing our paper, your suggestions are highly valuable to us. As the rebuttal phase is nearing its end, we want to kindly follow up to ensure there is sufficient time for any further clarifications or concerns you might have. Please let us know if there is anything further we can do to assist or elaborate on.
> >
> > Thank you once again for your time and effort in reviewing our paper.
> >
> > Best regards,
> >
> > The authors of submission 6951

---

> ### Author Response · Authors · 2024-11-27
> **Looking forward to receiving your feedback**
>
> Dear reviewer vQGg,
>
> We truly understand the large workload that comes with reviewing and deeply appreciate the effort and time you have dedicated to reviewing our paper.  As we are at the last stage of PDF updation, we want to kindly follow up to ensure there is sufficient time to address any remaining concerns you might have. Your recognition is highly important to us, and we sincerely hope to address all your concerns.
>
> Thank you once again for your efforts in reviewing our paper, and we look forward to receiving your feedback.
>
> Best regards,
>
> The authors of submission 6951

---

> ### Comment · Reviewer_vQGg · 2024-12-02
>
> I went through the entire manuscript over the weekend and I think this is fairly well written work.
> One last comment is to be more informative with figure captions, and enlarge the text in the figures (e.g., axis legend) for better visibility.

---

> > ### Author Response · Authors · 2024-12-02
> > **Thank you so much!**
> >
> > Thank you very much for raising your rating. We sincerely appreciate your recognition of our work and the valuable suggestions you provided, which have significantly helped us improve our work.
> >
> > Following your comments, we will revise the figure captions in the final manuscript to provide more detailed information and ensure that the text in the figures is enlarged for better readability.
> >
> > Thank you so much!

---

### Author Response · Authors · 2024-12-01
**Summary of Revisions**

We sincerely appreciate the valuable feedback from all reviewers, which has significantly improved our work. In response, we have uploaded a revised version of the paper with substantial updates to address all the concerns and suggestions. Below, we provide a summary of the revisions made:

---

**1. Baseline addition**: Based on the comments from reviewer 75RC and reviewer drzD, we have added 5 more baselines, including 2 typical machine learning methods and 3 state-of-the-art graph transformer methods.

---

**2. Ablation study**: We have merged node importance encoding ablation and functional module-aware self-attention ablation into Section 4.3. Figure 3 has been updated. The comparison between NE and other node importance measurement methods has been moved to Section 4.4.

---

**3. Model parameter comparison**: We have presented the number of parameters for different models on three datasets (Table 4).

---

**4. Biological plausibility in node importance encoding**: To enhance the evaluation of biological plausibility, we have compared NE value with both average functional connectivity strength (PCC value) and node efficiency value. Comparative curves across different node numbers for all datasets are provided in Appendix D.3 and D.4.

---

**5. Biological plausibility in functional module-aware self-attention**: We’ve revised Figure 5 by displaying empirical labels of brain regions and functional modules. It shows that the learned attention scores of our model align with the division of functional modules better than other methods. We have added Table 9 in Appendix D.5 for the detailed functional module division. To further analyse the relationships between the attention patterns learned by our model and the biological patterns of target diseases, we have visualized and compared the self-attention scores of samples from each class in the three datasets (Appendix D.6).

---

**6. Paragraph modification and section reorganisation**: We have thoroughly reviewed the manuscript and revised misleading or unclear sentences and paragraphs. Section ‘Model Generalizability Analysis’ has been moved to Appendix D.7.


---

We believe these updates address the reviewers' concerns and substantially improve the quality of the paper. We thank all reviewers for their constructive feedback and support.

---

### Meta-Review · Area_Chair_QXqQ · 2024-12-20

**Metareview:**

The paper presents BioBGT, a biologically informed graph transformer framework for brain graph representation learning. The method incorporates network entanglement-based node importance encoding and functional module-aware self-attention to capture brain network properties, demonstrating improved performance in brain disease detection tasks across several neuroimaging datasets.

The biological plausibility of the model is not sufficiently supported, both theoretically and experimentally. While the authors provide a compelling approach using node importance encoding and self-attention, the novelty of these methods is not clear, and more comparisons with traditional ML models like random forests or SVM would strengthen the paper. Additionally, the biological relevance of the model's outputs, particularly in relation to known brain structures, remains unsubstantiated. Most were addressed by the authors.

Despite some weaknesses, the paper is well-written, clear, and presents an innovative approach by incorporating concepts from quantum computing into neuroimaging. The model's performance is strong, with clear improvements over baseline methods, and the use of multiple neuroimaging datasets demonstrates its robustness. The authors also engage in detailed discussions of their methodology, and the experimental results are promising, especially in disease detection tasks.

**Additional Comments On Reviewer Discussion:**

During the rebuttal period, the authors addressed several key concerns raised by Reviewer vQGg, Reviewer 75RC, and Reviewer drzD, making clarifications and revisions to enhance the manuscript.

The authors clarified the novelty of their methods, specifically the encoding of node importance into representations, distinguishing it from simple measurement approaches. They refined the concept of functional module-aware self-attention and provided experimental comparisons to support the claims.

The authors elaborated on the use of quantum entanglement, particularly through the density matrix, to quantify structural information in brain networks. They emphasized its value in capturing correlations and information diffusion.

The authors revised their claims regarding the biological plausibility of their methods, particularly in relation to PCC values. They aligned their statements with current neuroscience knowledge and toned down certain claims, providing supporting references.

The authors improved Figure 5 and its caption, offering a more detailed analysis to clarify the relationship between functional modules and self-attention patterns, addressing Reviewer vQGg's concerns.

The authors addressed the lack of experiments with smaller parcellations and the absence of an ablation study for the community contrastive strategy. They discussed parcellation details for various datasets and clarified that ablation studies had been conducted.

The authors introduced baseline comparisons with traditional machine learning models (Random Forests and SVM), sharing experimental results and addressing concerns about hyperparameters.

In response to Reviewer drzD, the authors revised Figure 5 to include functional module labels and better align the identified modules with known brain regions, reinforcing their biological plausibility claim.

The authors revised their discussion on BioBGT’s generalizability to non-biological networks, clarifying that its primary focus is on biological networks and moving the section to the appendix.

Among all the improvements, the extended detailed explanation of biological plausibility is particularly appreciated and played a significant role in the final decision. While some concerns remained, the revisions were largely well-received, enhancing the manuscript's clarity, scientific rigor, and biological relevance.

---

### Decision · Program_Chairs · 2025-01-22

Accept (Poster)